# Probabilistic cost modeling as a basis for optimizing inspection and maintenance of turbine support structures in offshore wind farms

Muhammad Farhan[a], Ronald Schneider[a], Sebastian Thöns[a,b], Max Gündel[c]

[a] *Bundesanstalt für Materialforschung und -prüfung (BAM), Berlin, Germany*
[b] *Division of Structural Engineering, Lund University, Sweden*
[c] *Chair of Steel Construction, Helmut-Schmidt-Universität / Universität der Bundeswehr Hamburg, Germany*

**Correspondence to**: Muhammad Farhan (muhammad.farhan@bam.de)

## Abstract

The operational management of offshore wind farms includes inspection and maintenance (I&M) of the wind turbine support structures. These activities are complex and influenced by numerous uncertain factors that affect their costs. The uncertainty in the I&M costs should be considered in decision value analyses performed to optimize I&M strategies for the turbine support structures. In this paper, we formulate a probabilistic parametric model to describe I&M costs for the common case in which a wind farm is serviced and maintained using a workboat-based strategy. The model is developed based on (a) interviews with a wind farm operator, engineering consultants, and operation and maintenance engineers, as well as (b) on scientific literature. Our methodology involves deriving the probabilistic models of the cost model parameters based on intervals representing the subjective expert opinion on the foreseeable ranges of the parameter values. The probabilistic cost model is applied to evaluate the total I&M costs, and a sensitivity analysis is conducted to identify the main cost drivers. The model can be utilized to optimize I&M strategies at the component, structural system, and wind farm level. To illustrate its potential use, we apply it in a numerical study in which we optimize I&M strategies at the structural system level and identify and demonstrate a simplified approach of capturing uncertain I&M costs in the optimization. The simplified approach is generalized and made available for maintenance cost optimization of offshore wind turbine structures.

**Keywords**: offshore wind farms, turbine support structures, inspection and maintenance, probabilistic cost modeling, sensitivity analysis, decision value analysis

# 1  Introduction

The harsh offshore environment in combination with the rotor dynamics affect the condition and integrity of the wind turbine (WT) support structures in offshore wind farms. To improve the condition of deteriorated structural components and, consequently, to prevent failures of the WT support structures, wind farm operators perform condition-based or possibly predictive maintenance. Maintenance is classified as condition-based when scheduled based on the current component or system condition which is inferred from inspection/monitoring outcomes, and predictive when carried out based on the predicted component or system condition where the model-based predictions are informed by the available inspection/monitoring outcomes and the previously performed maintenance actions.

Wind farm operators typically conduct separate inspection and maintenance (I&M) campaigns to first collect information on the condition of the deteriorating WT support structures and to subsequently improve it if necessary. Within this context, an inspection campaign is characterized by the campaign time and inspection method, the number of inspected WT support structures, and the number and location of the inspected components in a WT support structure. In case the wind farm is serviced by boats operating from a port base, an inspection campaign is additionally influenced by the distance from the port to the wind farm, the choice of vessel, the number of personnel, the required equipment, the mobilization/demobilization activities, and the time to complete an inspection work package. Like inspection campaigns, a maintenance campaign is characterized by the campaign time and maintenance method. In addition, a maintenance campaign is influenced by the component location, the time to complete a maintenance intervention on site, the required equipment, preparations and materials, the distance between the port and the wind farm, the choice of vessel, the number of personnel, and the effort involved in engineering a maintenance intervention (i.e., designing and testing of a maintenance solution).

Clearly, I&M of deteriorating WT support structures in an offshore wind farm is associated with costs and the total lifetime I&M costs depend on the adopted I&M strategy, which determines the time and scope of each I&M campaign based on the available system information. Condition-based and predictive maintenance strategies can be optimized at the beginning of and adapted during the planned and/or extended lifetime of an offshore wind farm using preposterior analysis from Bayesian decision theory (e.g., Sorensen, 2009; Nielsen and Sorensen, 2011; Florian and Sorensen, 2017; Farhan, Schneider and Thöns, 2021; Bismut and Straub, 2021). In such an analysis, probabilistic models of (a) the governing deterioration processes including the effect of maintenance, (b) the structural performance, and (c) the inspection/monitoring performance are employed to predict:

- the condition of the structural components, inspection/monitoring outcomes and maintenance actions, and

- the structural component/system reliability conditional on the predicted component condition, inspection/monitoring outcomes, and maintenance actions.

In addition, a cost model is utilized to quantify the costs of inspections/monitoring and maintenance as well as the monetarized consequences of structural failures. Based on these models, the expected lifetime I&M costs and the lifetime risk of structural failure can be estimated for a given I&M strategy. A cost and risk optimal I&M strategy then balances the expected lifetime I&M costs with the lifetime risk of structural failure.

In the existing literature, normalized cost ratios or deterministic cost models are utilized as a basis for optimizing I&M of deteriorating structural systems using decision-theoretical approaches (e.g., Schneider, Rogge, Thöns et al., 2018; Bismut and Straub, 2021; Morato, Andriotis, Papakonstantinou et al., 2023). Although deterministic cost models enable an

optimization of I&M activities, they lack the ability to capture the effect of the I&M cost uncertainties in the decision analysis; especially in applications in which I&M costs are included in the underlying models on a non-linear basis. Importantly, probabilistic parametric cost modeling facilitates sensitivity analyses (beyond local derivative-based sensitivity analyses) to understand the effect of the various uncertain cost-affecting factors on the total I&M costs. With regards to optimizing I&M of WT support structures in offshore wind farms, comprehensive and explicit consideration of probabilistic I&M costs in the decision analysis is – to the best of the authors' knowledge – something which has not been explored previously. In addition, this issue is also relevant, since – from our experience – wind farm operators commonly highlight the need to consider the uncertainties in the I&M costs in the optimization of I&M strategies.

Motivated by this, we develop a probabilistic parametric cost model of I&M of WT support structures in offshore wind farms. In particular, we focus on wind farms which are serviced and maintained using a workboat-based strategy, where the workboats operate from a port base. The cost model is derived based on interviews with a wind farm operator, engineering consultants, and operation and maintenance engineers, as well as on scientific literature. Subsequently, we employ the model to (a) quantify the uncertainties in the total I&M costs and (b) perform a variance-based sensitivity analysis to better understand the key cost drivers. The model can be applied to optimize I&M at the component, structural system, and wind farm level. To demonstrate a potential application, we apply the cost model in a cost and risk-informed decision value analysis to optimize I&M strategy for a steel frame subject to fatigue.

The paper is organized as follows: Section 2 presents a generic decision-theoretical framework, which forms the basis for optimizing I&M of WT support structures in offshore wind farms. The presentation of the material follows our previous work (Farhan, Schneider and Thöns, 2021). However, compared to our previous work, our current contribution explicitly considers the uncertainties in the I&M costs in the decision analysis. In Section 3, different types of I&M methods for support structures in offshore wind farms are discussed. Subsequently, different uncertain parameters are identified that influence the overall costs of an I&M strategy. Furthermore, ranges of these parameters are also presented, as estimated based on the expert knowledge. To quantify the overall cost of the considered I&M activities, a deterministic cost model is first formulated in Section 4.1. Subsequently, in Section 4.2 a probabilistic I&M model is constructed by combining the deterministic cost model with a probabilistic model of its uncertain parameters. Section 5 presents the uncertainty quantification and sensitivity analysis. Section 6 summarizes the numerical example. A summary and concluding remarks are provided at the end of the paper in Section 7.

## 2 Utility-informed optimization of I&M of turbine support structures in offshore wind farms considering probabilistic I&M costs

### 2.1 Decision analysis

The identification of an optimal I&M strategy for turbine support structures in offshore wind farms is a decision problem under uncertainty and risk (Farhan, Schneider and Thöns, 2021). This type of problem can be solved based on Bayesian decision theory (Raiffa and Schlaifer, 1961) and graphically represented by the generic decision tree shown in Figure 1. Each branch of the decision tree corresponds to a realization of decisions represented by square nodes and random events/random variables represented by circular nodes. As an example, the lower branch in the decision tree in Figure 1 corresponds to a realization of (a) a decision $i$ concerning the inspection/monitoring regime, (b) the corresponding probabilistic inspection/monitoring outcomes $\mathbf{Z}_i$, (c) the decisions $\mathbf{a}$ concerning the maintenance actions, (d) the probabilistic

parameters **Y** influencing the effect of the maintenance actions **a**, (e) the probabilistic parameters**X** influencing the system state and (f) the probabilistic parameters **W** influencing the utility. Each of these realizations is associated with a utility $U$ and an index representing the analysis type, depicted with the diamond shaped leaves of the decision tree. According to utility theory (Von Neumann and Morgenstern, 1947), the maximization of the expected value of the utility $U$ quantifies the optimality of decisions. From this it follows that the optimal decisions concerning inspection/monitoring and maintenance can be determined by maximizing the expected utility.

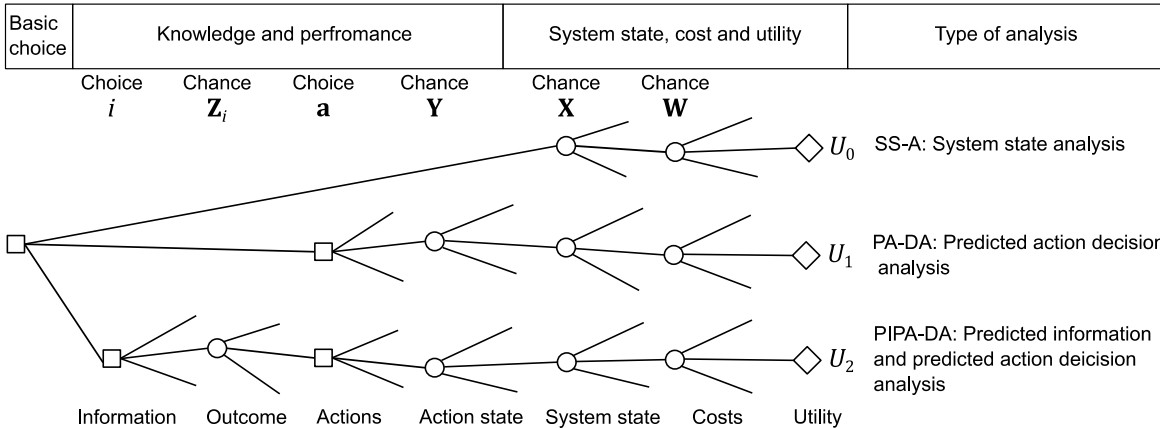

***Figure 1:*** *Generic decision tree for modeling decisions on inspection/monitoring and maintenance of turbine support structures in offshore wind farms (adapted from Thöns, 2018; Farhan, Schneider and Thöns, 2021) The tree consists of square nodes representing decisions, circular nodes representing random events/random variables and diamond shaped nodes representing utility.*

The utility $U_0$ associated with the upper branch of the decision tree can be described by the following generic utility function:

$$U_0(\mathbf{X}, \mathbf{W}) = B(\mathbf{X}, \mathbf{W}) - C_F(\mathbf{X}, \mathbf{W}) \tag{1}$$

where $B(\mathbf{X}, \mathbf{W})$ and $C_F(\mathbf{X}, \mathbf{W})$ describe the lifetime economic benefits from operating the wind farm and lifetime costs of structural failures in function of **X** and **W**. Based on the utility function $U_0(\mathbf{X}, \mathbf{W})$ and the prior probability distributions of **X** and **W**, the expected utility $\mathbb{E}[U_0]$ can be computed as described in Appendix A.1. This quantity is primarily required as a reference value in the decision value analysis presented below in Section 2.2. In the literature, the evaluation of $\mathbb{E}[U_0]$ is referred to as system state analysis (SS-A) (Thöns and Kapoor, 2019).

Next, the utility $U_1$ associated with the center branch of the decision tree can be generically expressed by the following utility function:

$$U_1(\mathbf{a}, \mathbf{Y}, \mathbf{X}, \mathbf{W}) = B(\mathbf{a}, \mathbf{Y}, \mathbf{X}, \mathbf{W}) - C_M(\mathbf{a}, \mathbf{Y}, \mathbf{X}, \mathbf{W}) - C_F(\mathbf{a}, \mathbf{Y}, \mathbf{X}, \mathbf{W}) \tag{2}$$

where $B(\mathbf{a}, \mathbf{Y}, \mathbf{X}, \mathbf{W})$, $C_M(\mathbf{a}, \mathbf{Y}, \mathbf{X}, \mathbf{W})$ and $C_F(\mathbf{a}, \mathbf{Y}, \mathbf{X}, \mathbf{W})$ are the lifetime economic benefits, maintenance costs and failure costs in function of **a**, **Y**, **X** and **W**. The optimal maintenance actions $\mathbf{a}^*$ are identified by maximizing the conditional expected utility $\mathbb{E}[U_1|\mathbf{a}]$, which is computed based on the utility function $U_1(\mathbf{a}, \mathbf{Y}, \mathbf{X}, \mathbf{W})$ and the prior probability distributions of **Y**, **X** and **W,** as outlined in Appendix A.2. Identifying $\mathbf{a}^*$ in this way (i.e., without considering inspection/monitoring to inform decisions on maintenance actions) is referred to as prior decision analysis (Raiffa and Schlaifer, 1961) or predicted action decision analysis (PA-DA) (see also Thöns and Kapoor, 2019).

Finally, the utility $U_2$ associated with the lower branch of the decision tree can be generically written as (see also Sorensen, 2009):

$$
\begin{aligned}
U_2(i, \mathbf{Z}_i, \mathbf{a}, \mathbf{Y}, \mathbf{X}, \mathbf{W}) \\
= B(i, \mathbf{Z}_i, \mathbf{a}, \mathbf{Y}, \mathbf{X}, \mathbf{W}) - C_{SHM}(i, \mathbf{Z}_i, \mathbf{a}, \mathbf{Y}, \mathbf{X}, \mathbf{W}) - C_I(i, \mathbf{Z}_i, \mathbf{a}, \mathbf{Y}, \mathbf{X}, \mathbf{W}) \\
- C_M(i, \mathbf{Z}_i, \mathbf{a}, \mathbf{Y}, \mathbf{X}, \mathbf{W}) - C_F(i, \mathbf{Z}_i, \mathbf{a}, \mathbf{Y}, \mathbf{X}, \mathbf{W})
\end{aligned}
\tag{3}
$$

where $B(i, \mathbf{Z}_i, \mathbf{a}, \mathbf{Y}, \mathbf{X}, \mathbf{W})$, $C_{SHM}(i, \mathbf{Z}_i, \mathbf{a}, \mathbf{Y}, \mathbf{X}, \mathbf{W})$, $C_I(i, \mathbf{Z}_i, \mathbf{a}, \mathbf{Y}, \mathbf{X}, \mathbf{W})$, $C_M(i, \mathbf{Z}_i, \mathbf{a}, \mathbf{Y}, \mathbf{X}, \mathbf{W})$ and $C_F(i, \mathbf{Z}_i, \mathbf{a}, \mathbf{Y}, \mathbf{X}, \mathbf{W})$ are the lifetime economic benefits, monitoring costs, inspection costs, maintenance costs and failure costs in function of $i$, $\mathbf{Z}_i$, $\mathbf{a}$, $\mathbf{Y}$, $\mathbf{X}$ and $\mathbf{W}$. The optimal information acquisition regime (or inspection/monitoring regime) $i^*$ is determined by maximizing the conditional expected utility $\mathbb{E}[U_2|i]$, which is determined based on $U_2(i, \mathbf{Z}_i, \mathbf{a}, \mathbf{Y}, \mathbf{X}, \mathbf{W})$ and the probabilistic models of $\mathbf{Z}_i$, $\mathbf{Y}$, $\mathbf{X}$ and $\mathbf{W}$ as detailed in Appendix A.3. As discussed in Appendix A.3, this maximization jointly optimizes decisions on inspection/monitoring and maintenance actions based on (a) predicted information on the system condition and performance, (b) predicted maintenance actions and (c) corresponding benefits and costs. This analysis is referred to as preposterior decision analysis (Raiffa and Schlaifer, 1961) or predicted information and predicted action decision analysis (PIPA-DA) (see also Thöns and Kapoor, 2019).

From Eq. (2) and (3) it can be seen that a model for quantifying the inspection/monitoring and maintenance costs is a key component of the utility functions required to measure the optimality of decisions concerning I&M of WT support structures in offshore wind farms. Such a cost model should be formulated in function of the uncertain parameters influencing the I&M costs in order to capture the uncertainties in these costs in the decision-making.

## 2.2 Decision value analysis

The root node in the decision tree in Figure 1 represents the basic decision concerning the implementation of an integrity management strategy (Thöns, 2018). This decision can be informed by a decision value (DV) analysis. Following Thöns and Kapoor (2019), three different DV may be formulated based on the decision tree shown in Figure 1. The first DV, $V_{PA-DA}^{PIPA-DA}$, is defined as the difference between the maximum expected utility resulting from the PIPA-DA, $\mathbb{E}[U_2|i^*]$, and the maximum expected utility resulting the PA-DA, $\mathbb{E}[U_1|\mathbf{a}^*]$, i.e.:

$$
V_{PA-DA}^{PIPA-DA} = \mathbb{E}[U_2|i^*] - \mathbb{E}[U_1|\mathbf{a}^*]
\tag{4}
$$

where $\mathbb{E}[U_2|i^*]$ and $\mathbb{E}[U_1|\mathbf{a}^*]$ are determined as described in Appendix A.

The second DV, namely the predicted value of information and actions $V_{SS-A}^{PIPA-DA}$, is defined as the difference between the maximum expected utility resulting from the PIPA-DA, $\mathbb{E}[U_2|i^*]$, and the expected utility resulting from the SS-A, $\mathbb{E}[U_0]$:

$$
V_{SS-A}^{PIPA-DA} = \mathbb{E}[U_2|i^*] - \mathbb{E}[U_0]
\tag{5}
$$

The third DV, i.e., the predicted value of actions $V_{SS-A}^{PA-DA}$, is the difference between the maximum expected utility resulting from PA-DA and the expected utility provided by the SS-A given as:

$$
V_{SS-A}^{PA-DA} = \mathbb{E}[U_1|\mathbf{a}^*] - \mathbb{E}[U_0]
\tag{6}
$$

Essentially, an integrity management strategy should be implemented if the value of $V_{PA-DA}^{PIPA-DA}$, $V_{SS-A}^{PIPA-DA}$ or $V_{SS-A}^{PA-DA}$ is positive.

# 3 I&M of turbine support structures in offshore wind farms

## 3.1 I&M methods

Inspections are performed to obtain information on the condition of structural components. Concerning offshore wind turbine support structures, they look for (indicators of) deterioration (e.g., corrosion and/or fatigue cracks) which has an effect on the integrity of the structural systems. In this contribution, a probabilistic cost model is developed for I&M actions performed to detect and repair fatigue cracks in welded connections in steel support structures of wind turbines in offshore wind farms (e.g., monopiles, jackets, etc.). In such structural systems, the welded components can be located above and below water level. The components located above water are typically part of the turbine tower, transition piece, main access platform, and access systems, which can be inspected via rope access and getting closer to the structure, while in the areas of the transition piece and sub structure below water, inspections are carried out by a diver, or by utilizing a remotely operated vehicle (ROV). The location of the inspected welded component has thus an effect on the required personnel, vessels, equipment, and logistics.

Two types of inspection methods to identify fatigue damage in welded components located above and/or below water level are considered: visual inspection, and electromagnetic (EM) inspection methods such as eddy current (EC), magnetic particle inspection (MPI) and alternating current field measurement (ACFM). Visual inspection is a coarse method capable of detecting only relatively large surface breaking defects in welds or fatigue failures of welded connections. It can be performed with the help of a camera mounted on an ROV or by naked-eye observation. In contrast, EM inspection methods detect smaller surface breaking defects in welds. They can also be applied by a diver below water.

After an inspection campaign, if any fatigue damage is detected, a subsequent maintenance action (e.g., a repair) is performed based on the inspection outcomes. Depending on the criticality of the identified fatigue damage, the maintenance campaign is launched in the same year or the following year.

During an inspection campaign, the length and depth of a detected surface-breaking defect is measured to inform decisions on the repair methods. With regards to possible repairs for welded joints, we consider two methods. The first repair method is referred to as welding (Rodriguez-Sanchez, Rodriguez-Castellanos, Perez-Guerrero et al., 2011). In this method, the welded joint is repaired by removing a surface crack through grinding and subsequent filling of the resulting groove with wet welding. This method is applied if the measured depth of the surface crack is greater than a defined percentage of the section thickness. Any surface crack with a measured depth less than the defined percentage of the section thickness may be repaired by grinding, which is the second repair method (Rodriguez-Sanchez, Dover and Brennan, 2004).

## 3.2 Factors affecting I&M costs

There are several factors that influence the total cost of inspecting and maintaining support structures in an offshore wind farm. In this contribution, the influencing factors are identified, and their costs are estimated based on the existing scientific literature and interviews with I&M experts: a wind farm operator, engineering consultants, and operation and maintenance engineers – whose expertise does not necessarily cover all types of offshore wind farms. These experts are, however, able to provide sufficiently detailed information on established operational procedures and approximate estimates of I&M costs. The precise figures of I&M costs always depend on the actual wind farm type and the existing operational constraints. In this study, we consider the common case in which the wind farm is serviced by workboats operating from a nearby port base. Therefore, other forms of wind farm access such as helicopters, are not considered in this contribution.

Accessibility is the main factor influencing the cost of I&M of turbine support structures in offshore wind farms. Depending on the location of the offshore wind farm, the type and location of the I&M activity (above water level (AW) and/or below water level (BW)), and the duration of the I&M activity, a certain type of vessel is employed. The choice of vessel for port-based operations could be a crew transfer vessel (CTV) or a service operation vessel (SOV). CTV are usually used for frequent operations and are generally small aluminum catamarans employed to transfer personnel to and from offshore sites on a daily basis. CTV do not have sufficient dynamic positioning redundancies to keep still during rough sea conditions. Their carrying capacity is usually 12 crew members who do 12-hour shifts. SOV are larger vessels designed and equipped to be present for a longer duration at the offshore wind farm for subsea or extensive I&M operations. These vessels have a capacity of around 40 technicians and can perform 24-hour operations with multiple shifts (each shift is 12 hours), which means that they come back to port only approximately once every two weeks (Martinez-Luengo and Shafiee, 2019). Table 1 shows the range of mobilization and demobilization costs and costs per shift for CTV and SOV.

*Table 1: Estimates of vessel costs*

| | Type of vessel | |
| --- | --- | --- |
| Type of vessel cost | CTV | SOV |
| Mobilization / demobilization (€) | 2,000 - 20,000 | 15,000 – 80,000 |
| Vessel cost per shift (€/shift) | 1,000 - 15,000 | 10,000 – 50,000 |

The mobilization and demobilization costs of the vessels cover several aspects like the commuting time to the offshore wind farm, fuel consumption of the vessel, equipment and material costs as well as project management costs, which account for logistics organization and reporting. The vessel cost per shift captures the number of people involved in the operation, the personnel costs, and the operational cost of the vessel during the I&M activity.

I&M also include the additional effort for engineering the required repairs. This effort is associated with costs as summarized in Table 2. The engineering costs usually dependent on the type of repair. In the case of grinding, the extra cost of engineering and preparation entails the design of the repair, laboratory tests, etc. In the case of welding, the cost entails the design of the repair, chambers for underwater repair work if required, laboratory tests, etc. This additional cost is ideally incurred once during the service life of a wind farm because the type of hotspots/components is known; thus, if any repair is performed, the implementation of the repair has already been planned for the specific type of hotspot in the support structures.

*Table 2: Estimates of engineering costs*

| Type of repair | Engineering cost (€) |
| --- | --- |
| Grind repair | 5,000 – 35,000 |
| Weld repair | 10,000 – 100,000 |

The duration to complete a I&M activity is another factor that strongly influences the total I&M cost. It depends, for example, on the weather conditions, the experience of the personnel, the condition of the asset and the existence of marine growth. The total time to complete a I&M activity usually entails transit time between WT, the time required to complete the work package once stationed at a WT, and additional weather downtime due to unfavorable weather conditions. An increase in I&M activity time due to the aforementioned factors can lengthen

the offshore time within a campaign. While this may not seem crucial, the time increase has an effect on other costs, such as the costs of the deployment of a vessel, personnel, and equipment. Table 3 shows the estimation of the time that each of the I&M activities takes for a single component in a support structure. A component is here defined as a hotspot in a welded connection (i.e., a certain section of a weld).

*Table 3: I&M activity: type, location, and estimates of the duration per component*

| Type of activity | Location | hrs./component |
|---|---|---|
| Weld repair | Above water | 50 – 58 |
| | Below water | 60 – 70 |
| Grind repair | Above water | 14 – 18 |
| | Below water | 24 – 30 |
| Visual inspection | Above water | 1 – 2 |
| | Below water | 5 – 8 |
| EM inspection | Above water | 4 – 6 |
| | Below water | 10 – 15 |

Weather downtime is mostly dependent on the type of vessel utilized and is given in Table 4. In the case of CTV, the weather downtime is usually higher because they are small in size and lighter compared to SOV and can easily lose position, especially if there are large waves and strong currents, while SOV can safely withstand the harshest conditions even in winter.

*Table 4: Estimates weather downtime in function of the vessel type*

| Type of vessel | Weather downtime |
|---|---|
| CTV | 30 – 40% |
| SOV | 10 – 15 % |

The transit time between turbines also influences the inspection maintenance cost. It is here estimated based Martinez-Luengo and Shafiee (2019), and usually varies between 15 to 30 minutes.

# 4 Parametric model of I&M costs

## 4.1 Deterministic model

Based on the discussion on the factors affecting I&M costs in Section 3.2, we develop a deterministic parametric cost model to describe the total cost of I&M of WT support structures in an offshore wind farm. This cost can be generally broken down into a campaign cost $C_C$, engineering cost $C_E$, and operational cost $C_{Op}$. In case I&M are performed simultaneously (mixed I&M), the total cost $C_{I\&M}$ is given as:

$$C_{I\&M} = C_C + C_E + C_{Op} \qquad (7)$$

In the usual case where inspections and maintenance are performed in separate campaigns, the total cost of inspection $C_I$ and the total cost of maintenance $C_M$ is given by:

$$C_I = C_C + C_{I,Op}$$
$$C_M = C_C + C_E + C_{M,Op}$$

(8)

The campaign cost $C_C$ corresponds to the fixed one-time cost of initiating the I&M activities, which includes the cost of commuting to the wind farm and back, the cost of the equipment and materials required for the planned activities, fuel costs, and project management costs. These cost components are included in the mobilization and demobilization cost of the vessels (see Table 1). As discussed in Section 3.2, a CTV or SOV can be selected as a vessel for the planned I&M activities depending on their nature and extent. In the case of maintenance, an additional cost for planning and engineering repairs $C_E$ has to be considered. This cost depends on the chosen repair method (welding or grinding).

Moreover, $C_{I,Op}$ and $C_{M,Op}$ are the operational costs of inspection and maintenance, which are the costs of conducting the inspection or maintenance operation when the vessel is at the offshore wind farm. The total operational costs further depend on the time to complete the operation, the vessel type, and its shift pattern. The total time to complete the operation depends on the extent of the I&M activity and where it is carried out, i.e., above or below water. The operational cost of inspection $C_{I,Op}$ and maintenance $C_{M,Op}$ activity is given by:

$$C_{I,Op} = \frac{t_{I,Op}}{t_{\text{shift}}} \cdot C_{\text{shift}}$$

$$C_{M,Op} = \frac{t_{M,Op}}{t_{\text{shift}}} \cdot C_{\text{shift}}$$

(9)

where $C_{\text{shift}}$ is the cost of the vessel (CTV or SOV) per shift, $t_{\text{shift}}$ is the duration of a shift (in hours) (see Section 3.2), $t_{I,Op}$ is the total time (in hours) to complete the overall inspection operation, and $t_{M,Op}$ is the total time to complete the overall maintenance operation. The total operational time for I&M is estimated as:

$$t_{I,Op} = \left[ \left[ \sum_{i=1}^{n_{I,WT}} n_{I,C,i} \cdot t_{I,C} \right] + (n_{I,WT} - 1) \cdot t_{\text{transit}} \right] \cdot (1 + WD)$$

$$t_{M,Op} = \left[ \left[ \sum_{i=1}^{n_{M,WT}} n_{M,C,i} \cdot t_{M,C} \right] + (n_{M,WT} - 1) \cdot t_{\text{transit}} \right] \cdot (1 + WD)$$

(10)

where $n_{I,WT}$ is the total number of WT to be inspected in an offshore wind farm during an inspection campaign, $n_{M,WT}$ is the total number of WT to be repaired in the offshore wind farm during a maintenance campaign, $n_{I,C,i}$ is the number of components to be inspected in the $i$th WT support structure in the offshore wind farm, $n_{M,C,i}$ is the number of components to be repaired in the $i$th WT support structure in the offshore wind farm, $t_{\text{transit}}$ is the transit time between the different WT, $t_{I,C}$ is the time to inspect a component above or below the water, and $t_{M,C}$ is the time to repair a component above or below water. Furthermore, $WD$ is the weather-related downtime, which is here defined relative to the overall operation time. This parameter also depends on the vessel type.

## 4.2 Probabilistic model

The model developed in Section 4 provides deterministic estimates of the I&M costs. However, due to the uniqueness of each operation, and the complexity of the individual activities involved, the different parameters governing Eq. (7), (8), (9), and (10) – i.e., the campaign cost, the vessel costs, the engineering costs, the inspection duration, the repair duration, the transit time, and the weather downtime – are uncertain and their values are typically only known in terms of intervals, as estimated in Section 3.2. To capture these uncertainties, the parameters of the cost model are modeled as random variables. By probabilistically modeling the uncertainties in the parameters and propagating them through the deterministic cost model, a probabilistic description of the I&M costs is obtained.

Let **W** generically denote the vector of random variables influencing the total I&M costs. Based on Eq. (8), (9), and (10), the probabilistic cost model of inspection $C_I(\mathbf{W})$ can now be written as:

$$C_I(\mathbf{W}) = C_C + \frac{\left[\left[\sum_{i=1}^{n_{I,WT}} n_{I,C,i} \cdot t_{I,C}\right] + (n_{I,WT} - 1) \cdot t_{\text{transit}}\right] \cdot (1 + WD)}{t_{\text{shift}}} \cdot C_{\text{shift}} \tag{11}$$

Similarly, the probabilistic cost model of maintenance $C_M(\mathbf{W})$ is formulated as:

$$C_M(\mathbf{W}) = C_C + C_E + \frac{\left[\left[\sum_{i=1}^{n_{M,WT}} n_{M,C,i} \cdot t_{M,C}\right] + (n_{M,WT} - 1) \cdot t_{\text{transit}}\right] \cdot (1 + WD)}{t_{\text{shift}}} \cdot C_{\text{shift}} \tag{12}$$

Given the lack of empirical data on the uncertain parameters of the cost models **W**, their probabilistic models are – in a Bayesian sense – chosen based on the available expert knowledge. It should be emphasized upfront, however, that these probabilistic models can be updated using Bayesian methods if data on the parameters **W** become available.

As a first step in the probabilistic model building, the parameters in **W** are assumed to be independent and their marginal distributions are assumed to follow the lognormal distribution. The first assumption is made as no information on the correlation of the different parameters is available. The second assumption is supported by the following reasons. First, each parameter of the cost models only takes non-negative values, and their statistical distribution is typically expected to be unimodal, i.e., one range of values in the distribution occurs more frequently than other ranges of values. The lognormal distribution is commonly chosen to probabilistically model such quantities as it is bounded by zero, has no upper limit, and is unimodal. Second, the lognormal distribution is skewed to the right with a long tail capturing rare extreme values of the cost model parameters. Third, the assumption that the parameters are lognormal distributed can be partially explained by the central limit theorem. Assuming that each parameter of the cost model itself derives from a multiplicative process, the sum of the logarithms of the factors in the underlying process approaches a normal distribution and their product approaches a lognormal distribution as the number of factors becomes large. For these reasons, the lognormal distribution is a plausible probabilistic model for the different cost model parameters (see also Moy, Chen and Kao, 2015).

As a second step in the model building, the statistics of the different lognormal distributions are determined based on the lower and upper bounds of each cost model parameter specified in Section 3.2. These bounds represent the available expert knowledge on the ranges of the parameter values. Based on additional expert judgement, the lower and upper bound are assumed to characterize the 1% and 95%-quantile of the parameter values. Using this information, the different lognormal distributions are fitted as illustrated in Figure 2. The

resulting mean and coefficient of variation (CoV) of each probabilistic parameter of the cost models is summarized Table 5.

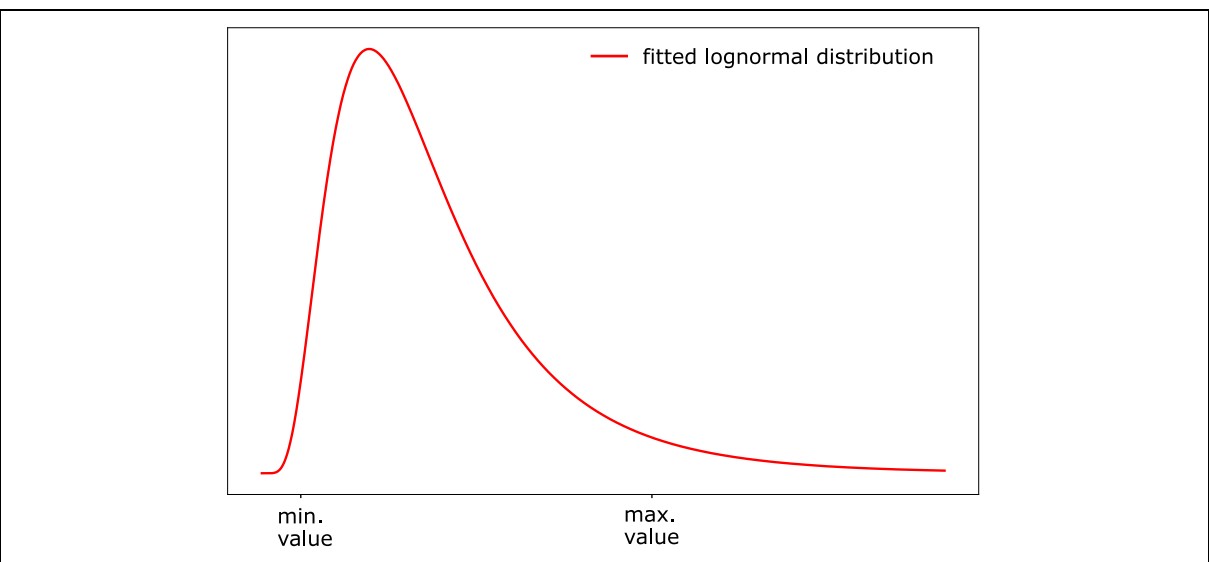

**Figure 2:** *Lognormal distribution fitted based on the lower and upper bound on the corresponding cost model parameter. The lower and upper bound represent the 1% and 95%-quantile of the parameter values.*

**Table 5:** *Mean and coefficient of variation (CoV) of the probabilistic parameters of the cost model*

| Parameter | Description | Unit | Distribution | Mean | CoV |
|-----------|-------------|------|--------------|------|-----|
| $C_{C,\text{CTV}}$ | Campaign cost (CTV) | [€] | lognormal | 9111.04 | 0.63 |
| $C_{C,\text{SOV}}$ | Campaign cost (SOV) | [€] | lognormal | 43771.22 | 0.44 |
| $C_{\text{shift,CTV}}$ | Vessel cost per shift (CTV) | [€/shift] | lognormal | 6220.67 | 0.78 |
| $C_{\text{shift,SOV}}$ | Vessel cost per shift (SOV) | [€/shift] | lognormal | 27744.47 | 0.42 |
| $t_{EM,BW}$ | Duration of component inspection (EM, below water) | [hrs.] | lognormal | 12.73 | 0.10 |
| $t_{EM,AW}$ | Duration of component inspection (EM, above water) | [hrs.] | lognormal | 5.10 | 0.10 |
| $t_{V,BW}$ | Duration of component inspection (visual inspection, below water) | [hrs.] | lognormal | 6.63 | 0.12 |
| $t_{V,AW}$ | Duration of component inspection (visual inspection, above water) | [hrs.] | lognormal | 1.53 | 0.17 |
| $C_{E,weld}$ | Engineering cost (welding) | [€] | lognormal | 45719.43 | 0.63 |
| $C_{E,grind}$ | Engineering cost (grinding) | [€] | lognormal | 17584.96 | 0.51 |
| $t_{weld,BW}$ | Duration of component repair (welding, below water) | [hrs.] | lognormal | 65.74 | 0.03 |
| $t_{weld,AW}$ | Duration of component repair (welding, above water) | [hrs.] | lognormal | 54.59 | 0.03 |

| | | | | | |
|---|---|---|---|---|---|
| $t_{grind,BW}$ | Duration of component repair (grinding, below water) | [hrs.] | lognormal | 27.41 | 0.05 |
| $t_{grind,AW}$ | Duration of component repair (grinding above water) | [hrs.] | lognormal | 16.24 | 0.06 |
| $t_{\text{transit}}$ | Transit time between turbines | [hrs.] | lognormal | 0.38 | 0.17 |
| $WD_{\text{CTV}}$ | Weather downtime (CTV) | - | lognormal | 0.35 | 0.07 |
| $WD_{\text{SOV}}$ | Weather downtime (SOV) | - | lognormal | 0.12 | 0.10 |

# 5   Quantification of uncertainties in I&M costs and sensitivity analysis

## 5.1   Uncertainty quantification

The probabilistic cost models for I&M of turbine support structures in the offshore wind farm defined in Equations (11) and (12) can be applied for different combinations of input parameters. The different combinations are defined by the vessel type, the inspection and repair methods, the number of inspected and/or repaired components above and/or below water level, and the number of inspected and/or repaired wind turbine support structures. For illustration, the probabilistic total I&M costs are in the following quantified at wind farm, wind turbine and component level.

First, the total I&M costs are estimated at wind farm level. To this end, it is assumed that $n_{I,C,BW} = 10$ components of $n_{I,WT} = 10$ support structures are inspected below water. In addition, it is assumed that $n_{M,C,BW} = 5$ components of $n_{M,WT} = 5$ support structures are repaired below water. Inspections are performed using EM and visual inspection methods, while welding and grinding are applied as repair methods. Moreover, a CTV is utilized as workboat in each scenario. For each considered scenario, the probabilistic distributions of the total I&M costs are determined using Monte Carlo (MC) simulations with $n_{MC} = 10^6$ samples of the corresponding model parameters. In the analysis, the model parameters are assumed to be statistically independent. The resulting empirical probability distributions of the total I&M costs are shown in Figure 3.

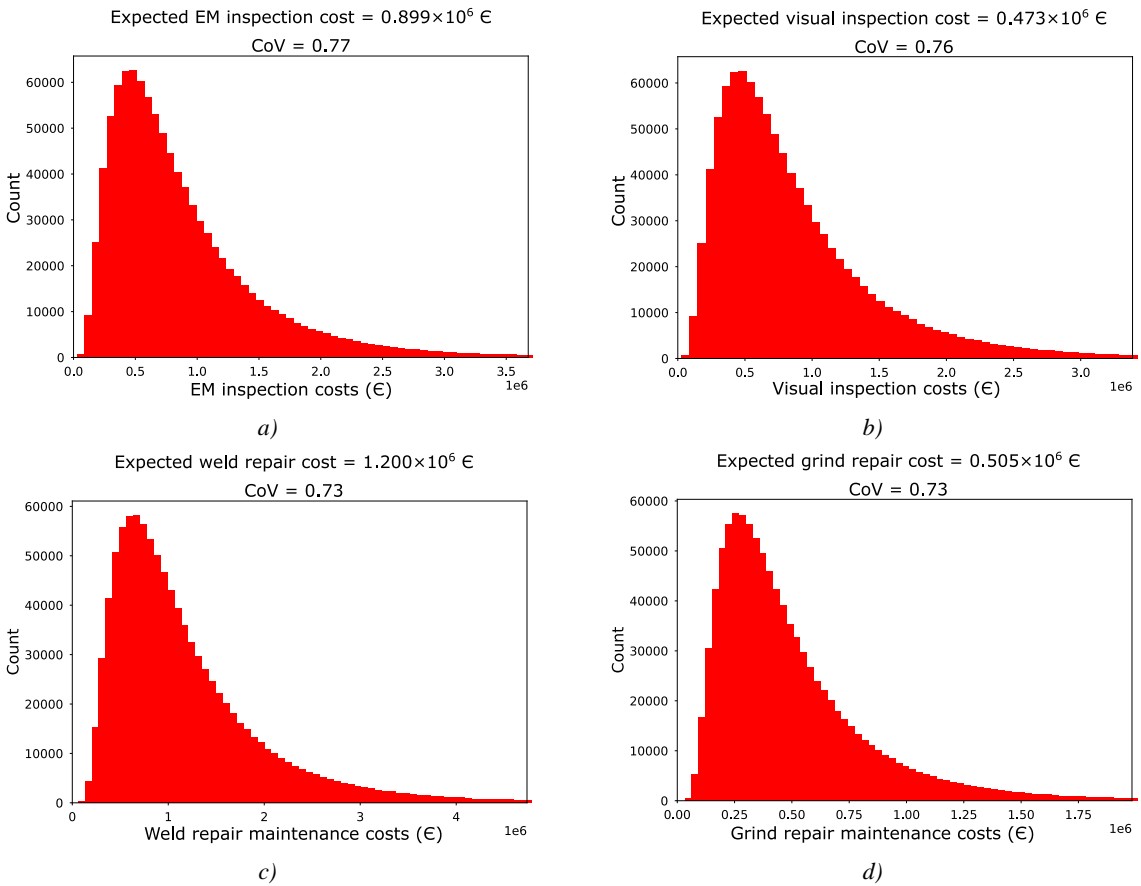

**Figure 3:** *Empirical probability distributions of the total I&M costs: a) 10 components in each 10 support structures are inspected below water with EM inspection technique, b) 10 components in each 10 support structures are visually inspected below water, c) 5 components in each 5 support structures are repaired below water by welding, d) 5 components in each 5 support structures are repaired below water by grinding*

Second, the total I&M costs are estimated at turbine level. In this case, it is assumed that 10 components of a support structure are inspected below water level, i.e., $n_{I,WT} = 1$, $n_{I,C,BW} = 10$; and 5 components in a support structure are repaired below water level, i.e., $n_{M,WT} = 1$, $n_{M,C,BW} = 5$. The assumptions regarding inspection and repair methods and the choice of vessel are the same as in the previous scenario considering I&M at wind farm level. The estimated empirical probability distributions of the total I&M costs together with their expected value and CoV are shown in Figure 4.

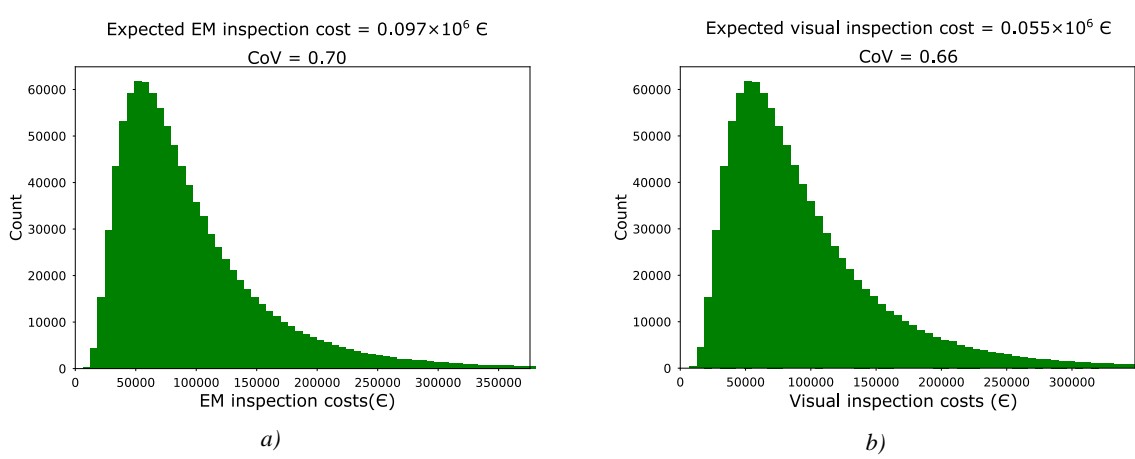

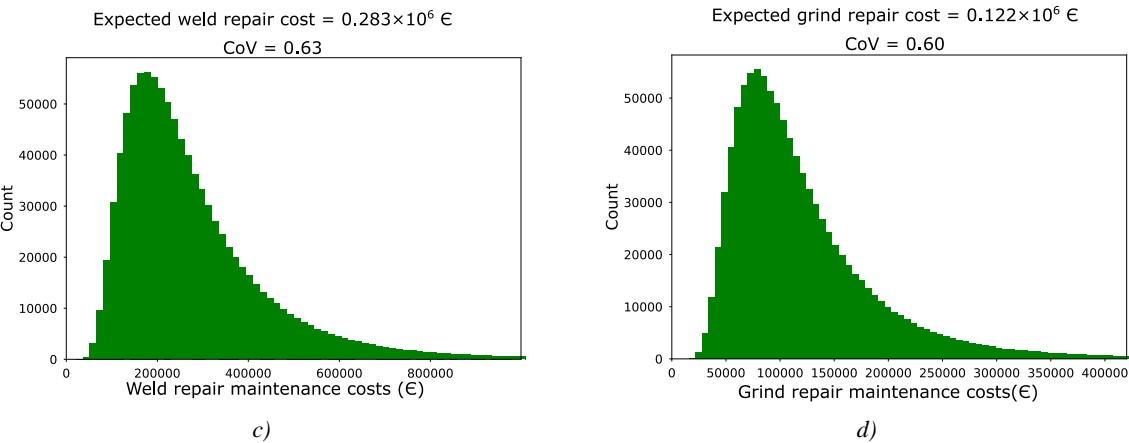

*c)*                                                                                          *d)*

**Figure 4:** *Empirical distributions of the total I&M costs: a) 10 components of a turbine support structures are inspected below water with EM inspection technique, b) 10 components of a turbine support structures are visually inspected below water, c) 5 components of a turbine support structures are repaired below water by welding, d) 5 components of a turbine support structures are repaired below water by welding*

Finally, the total I&M costs are estimated at the element level. In this scenario, $n_{I,C,BW} = 1$ components of $n_{I,WT} = 1$ turbines are inspected below water. The same is assumed for the maintenance campaign, i.e., only $n_{M,C,BW} = 1$ component of $n_{M,WT} = 1$ turbine support structure is repaired below water. The assumptions regarding inspection and repair methods and the choice of vessel are the same as in the scenario considering I&M at wind farm level. The empirical probability distributions of the total I&M costs are shown in Figure 5.

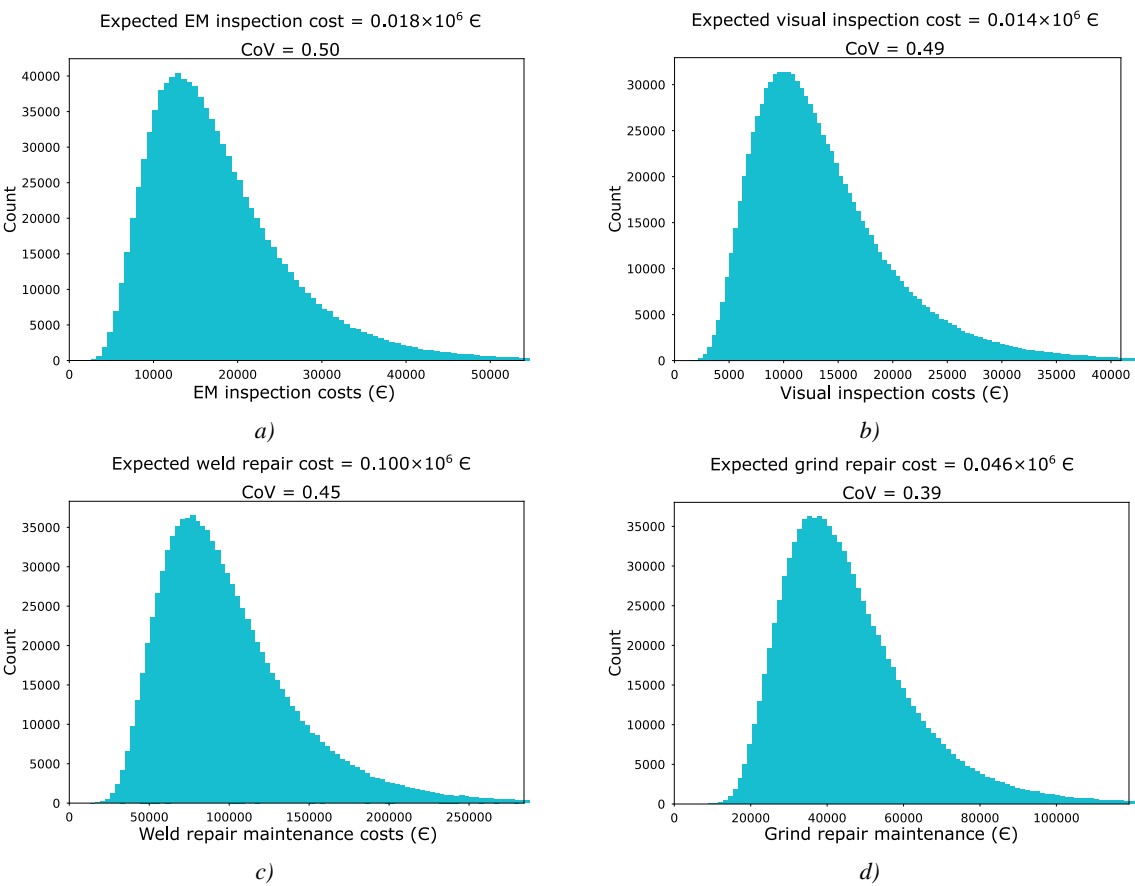

*c)*                                                                                          *d)*

**Figure 5:** *Empirical distributions of the total I&M costs: a) EM inspection of a component in a support structure below water level, b) visual inspection of a component in a support structure below water level, c) welding repair maintenance of a component repaired below water level for a support structure, d) grinding repair maintenance of a component repaired below water level for a support structure*

From Figure 3 to Figure 5, it can be seen that propagating the uncertainties in the cost model parameters through the cost models defined in Eq. (11) and (12) provides a probabilistic description of the total I&M costs. In each considered scenario, the total I&M costs exhibit an approximate lognormal distribution. The statistics of the total I&M costs shown Figure 3 to Figure 5 are summarized in Table 6. Note that the coefficients of variation (CoV) of the total I&M costs indicate that in each scenario certain parameters dominate the uncertainty in the total I&M costs. As an example, in Figure 3a (wind farm level analysis considering EM inspection), the CoV of the total I&M costs is similar to the CoV of the vessel cost per shift. This finding is further substantiated by the variance-based sensitivity analysis in Section 5.2.1, where in Figure 6 we observe that the vessel cost per shift has a sensitivity index close to one for the same inspection scenario.

*Table 6: Summary of the statistics of the total I&M costs*

|  |  | **Wind farm** | **Wind turbine** | **Component** |
|---|---|---|---|---|
| **EM inspection** | Expected value ($10^6$ €) | 0.899 | 0.097 | 0.018 |
|  | CoV | 0.77 | 0.70 | 0.50 |
| **Visual inspection** | Expected value ($10^6$ €) | 0.473 | 0.055 | 0.014 |
|  | CoV | 0.76 | 0.66 | 0.49 |
| **Weld repair** | Expected value ($10^6$ €) | 1.200 | 0.283 | 0.100 |
|  | CoV | 0.73 | 0.63 | 0.45 |
| **Grind repair** | Expected value ($10^6$ €) | 0.505 | 0.122 | 0.046 |
|  | CoV | 0.73 | 0.60 | 0.39 |

## 5.2 Sensitivity analysis

As discussed in Sections 3 and 4, the total I&M costs are influenced by numerous uncertain parameters $\mathbf{W}$. To study the importance of each model parameter $W_i$, a variance-based sensitivity analysis is performed (Sobol, 1993), which quantifies $W_i$'s effect on the variance of the inspection and maintenance costs in terms of the following first-order measure:

$$V_i = \text{Var}_{W_i}\{\mathbb{E}_{\mathbf{W}_{-i}}[C(\mathbf{W})|W_i]\} \tag{13}$$

where $C(\mathbf{W})$ can be the probabilistic model of the inspection costs defined in Eq. (11) or the probabilistic model of the maintenance costs defined in Eq. (12), $\mathbb{E}_{\mathbf{W}_{-i}}[C(\mathbf{W})|W_i]$ is the expected value of the inspection or maintenance costs with respect to all parameters except $W_i$ whose value is fixed, and $\text{Var}_{W_i}\{\mathbb{E}_{\mathbf{W}_{-i}}[C(\mathbf{W})|W_i]\}$ is the variance of this average model. Normalizing $V_i$ with the variance $\text{Var}[C(\mathbf{W})]$ provides the first order sensitivity index $S_i$ (Sobol, 1993):

$$S_i = \frac{V_i}{\text{Var}[C(\mathbf{W})]} = \frac{\text{Var}_{W_i}\{\mathbb{E}_{\mathbf{W}_{-i}}[C(\mathbf{W})|W_i]\}}{\text{Var}[C(\mathbf{W})]} \tag{14}$$

in which $\text{Var}[C(\mathbf{W})]$ is the variance of the inspection or maintenance costs. $S_i$ is here evaluated using a MC approach (Sobol, 2001).

### 5.2.1 Inspection costs

The first part of the sensitivity study analyses the effect of the campaign cost $C_c$, vessel cost per shift $C_{\text{shift}}$, and inspection operation time $t_{I,Op}$ on the total inspection costs based on the probabilistic cost model defined in Eq. (11), where the inspection operation time depends on the duration of a component inspection, the transit time between turbines, and the weather downtime.

For the inspection campaign, we perform a sensitivity analysis assuming that inspections are performed using the EM inspection method via a CTV. The study explores various scenarios related to component location (above or below water), the number of inspected turbine support structures, and the quantity of inspected components within each structure. Figure 6 illustrates these scenarios, with columns representing the number of inspected turbines and rows corresponding to component location. Each subplot displays sensitivity indices for campaign cost, vessel operation cost, and inspection operation time as functions of the number of inspected components.

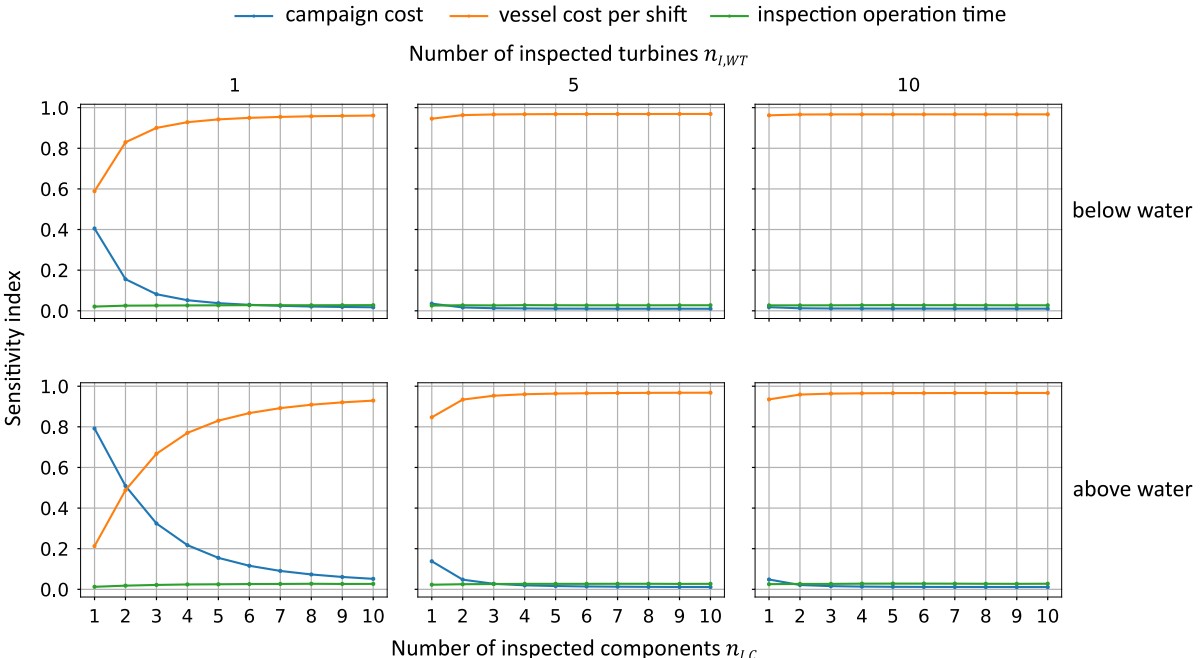

**Figure 6:** *Sensitivity indices of the campaign cost $C_c$, vessel cost per shift $C_{shift}$, and inspection operation time $t_{I,Op}$ in function of the location of the inspected components (above or below water), the number inspected turbine support structures and the number of inspected components in each support structure. Inspections are performed using the EM inspection method via a CTV.*

Figure 6 reveals that the vessel cost per shift exerts the most significant impact on total inspection cost throughout various scenarios, while the inspection operation time has the smallest effect. Campaign costs only become significant when a few components are inspected in one WT support structure. We obtain similar results when considering visual inspections as inspection method and an SOV as a workboat.

### 5.2.2 Maintenance costs

The second part of our sensitivity study evaluates the impact of the campaign cost $C_c$, the engineering costs $C_E$, the vessel cost per shift $C_{\text{shift}}$, and repair operation time $t_{M,Op}$ on the maintenance costs defined by Eq. (12). In the analysis, we assume that welding is used as repair method and a CTV is utilized as the workboat. The study considers various scenarios related to

the component location (above or below water), the number of repaired turbine support structures, and the number of repaired components within each support structure.

In Figure 7, we observe that at the turbine level the engineering costs have the greatest impact on the total maintenance costs, while the other parameters only have a small influence. Additionally, the influence of the vessel cost per shift increases with the number of repaired turbines and components, while the impact of the engineering costs decreases. Similar results are obtained when considering grinding as repair method and an SOV as a workboat.

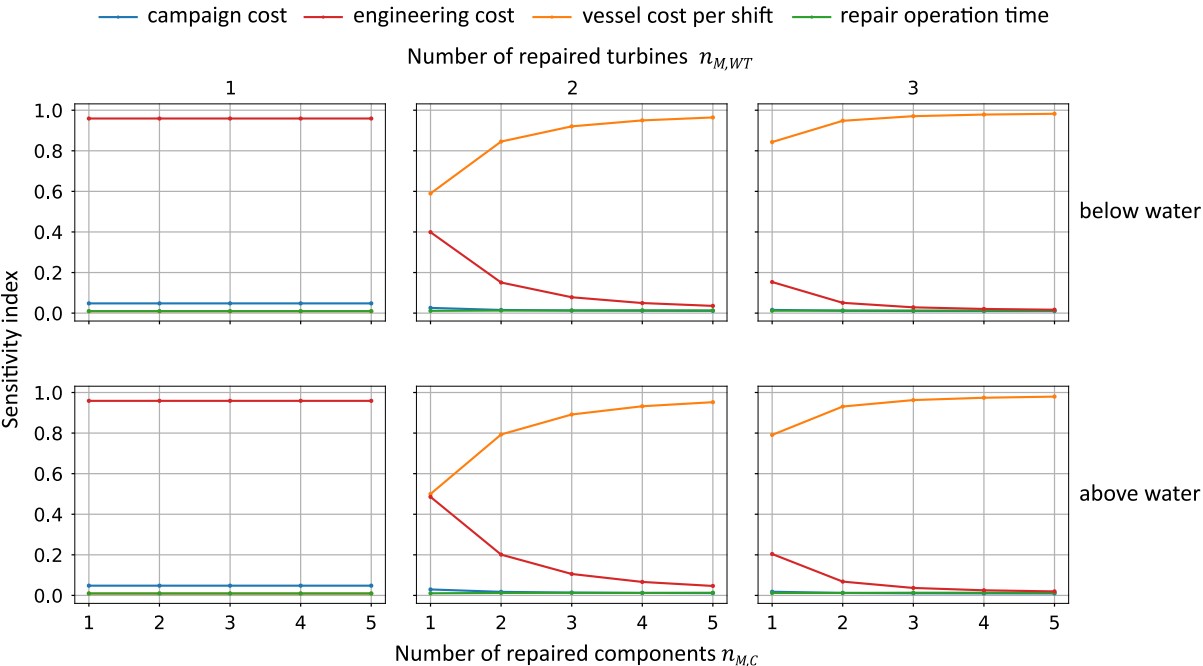

**Figure 7:** *Sensitivity indices of the campaign cost $C_C$, the cost of engineering repairs $C_E$, the vessel cost per shift $C_{shift}$, and the repair operation time $t_{M,Op}$ in function of the location of the repaired components (above or below water), the number repaired turbine support structures and the number of repaired components in each support structure. Repairs are performed using the welding repair method and a CTV.*

# 6   Numerical example

In the following, the probabilistic cost model formulated in Section 4 is applied in a cost and risk-informed optimization of an I&M strategy for the two-dimensional steel frame resembling a jacket support structure of an offshore wind turbine (see Figure 8). The optimization is performed at the beginning of its lifetime based on information from the design phase. The frame has been studied in numerous publications and in the following we provide a summary of the underlying models and assumptions. A more detailed description of the frame can be found in (Schneider, Thöns and Straub, 2017; Schneider, 2020; Eichner, Schneider and Baeßler, 2023).

The steel frame is made of welded tubular sections. Its planned lifetime is 25 years, which is divided into $j = 1, \ldots, m$ intervals of one year length. During the operational phase, the frame is exposed to a time-dependent lateral force representing a storm load. This load is modeled by its annual maximum $L_{max,j}$. In addition to storm loads, the frame – like a jacket support structure of an offshore wind turbine – is subject to fatigue due to dynamic excitations. In the current analysis, the welded connections of the frame contain 22 critical fatigue hotspots, which are indicated as red dots in Figure 8. The hotspot fatigue demand is quantified by the corresponding distributions of the fatigue stress ranges. Typically, these distributions are derived from an overall dynamic response analysis. In the current example, they are – as

described in Straub (2004) – determined based on the available design information (i.e. the hotspot fatigue lives and the applied SN curves).

Fatigue deterioration of the hotspots is described by probabilistic Paris-Erdogan fatigue crack growth models. The statistical dependence among the fatigue behavior of different hotspots is captured by introducing correlations among the uncertain parameters of the hotspot fatigue models. This correlation influences the system reliability and has an impact on the (optimal) inspection and maintenance regime.

The hotspots are inspected with MPI via a CTV and repaired by welding if required. The applied repair model is documented in detail in (Farhan, Schneider and Thöns, 2021). It is assumed that hotspots 1 to 8 are located above water, while hotspots 9 to 22 are located below water. The location of the of the hotspots (above or below water) influences the cost of inspections and repairs.

The time-dependent failure probability is computed by coupling the probabilistic fatigue deterioration models with a probabilistic structural performance model utilized to evaluate the system failure probability conditional on the hotspot condition. Inspection information is included in the estimation of the system failure probability through Bayesian updating of the probabilistic fatigue models. Further information regarding the applied fatigue, structural performance, and inspection models as well as the methods employed to compute the (updated) time-variant failure probability of the frame is documented in (Schneider, Thöns and Straub, 2017; Schneider, 2020; Eichner, Schneider and Baeßler, 2023).

In the current application, all consequences (costs of inspections and repairs as well as failure consequences) are expressed as monetary costs $C$ to facilitate quantitative DV analyses. Furthermore, as discussed in Nielsen and Sorensen (2021), the economic benefits from the existence of the wind turbine may be assumed to be independent of the structural reliability and I&M actions. In this case, they are constant and, consequently, can be neglected in the optimization of I&M of the WT support structure. It follows that the utility $U$ introduced in Section 2.1 is proportional to $-C$.

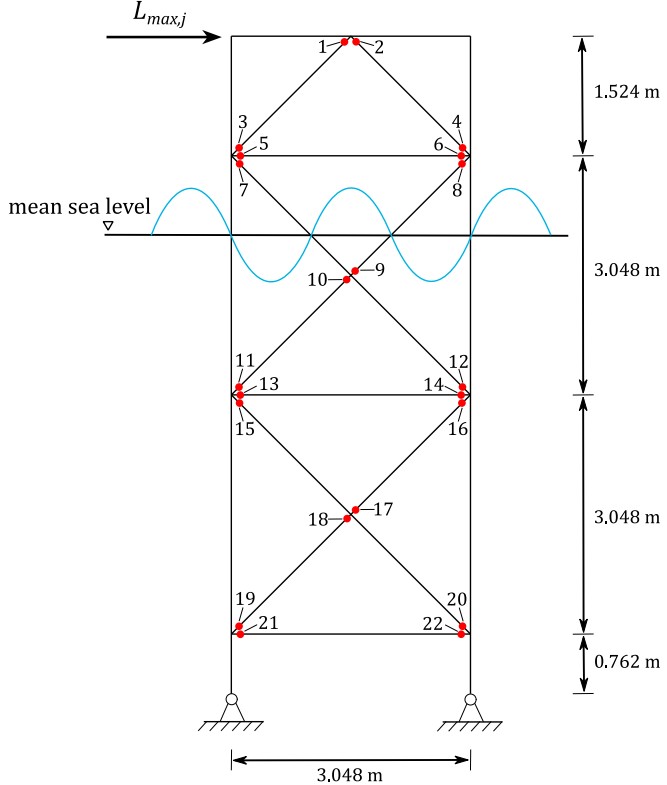

*Figure 8.: Steel frame with 22 fatigue hotspots indicated as red dots (adapted from Schneider, Thöns and Straub (2017))*

## 6.1 System state analysis

The SS-A determines the expected total lifetime cost $\mathbb{E}[C_0]$ assuming that no inspections and no maintenance actions are performed during the lifetime of the support structure. $\mathbb{E}[C_0]$ is equal to the expected total lifetime cost of system failure $\mathbb{E}[C_F]$ (lifetime risk of failure), i.e.:

$$\mathbb{E}[C_0] = \mathbb{E}[C_F] = \sum_{j=1}^{m} c_F \cdot \gamma_j \cdot [\Pr(F_j) - \Pr(F_{j-1})] \tag{15}$$

where $c_F = 2 \cdot 10^7 \text{€}$ is the failure cost, which is here assumed to be deterministic and equal to the investment cost of one wind turbine (Thöns, Faber and Val, 2017); $\Pr(F_j)$ is the cumulative probability of failure up to the end of year $j$; $\Pr(F_j) - \Pr(F_{j-1})$ is the probability of failure in year $j$; and $\gamma_j$ is the discounting function which discounts the failure cost $c_F$ occurring in year $j$ back to the present. The discounting function is defined as $\gamma_j = 1/(1+r)^j$, wherein $r = 0.02$ is the discount rate. The expected total lifetime cost $\mathbb{E}[C_0]$ of the case study related to steel frame is $7 \cdot 10^5 \text{€}$.

## 6.2 Predictive information and predictive action decision analysis considering probabilistic I&M costs

The PIPA-DA for jointly optimizing I&M is performed using the heuristic cost and risk-informed approach proposed in (Luque and Straub, 2019; Bismut and Straub, 2021), which corresponds in essence to the normal from of the preposterior decision analysis. We adopt this approach as it is computationally tractable compared to the extensive form of analysis described in Appendix A.3. In this approach, a I&M strategy, generically denoted by $S$, is defined by

parameterized rules that specify what, when, and how to inspect and repair based on the available system information (i.e., inspection outcomes and corresponding repairs as well as the predicted system failure probability conditional on the inspection outcomes and previously performed repairs). In the current application, the parameterized rules are defined as follows (see also Bismut, Luque and Straub, 2017; Eichner, Schneider and Baeßler, 2023; Schneider, 2019):

1. Inspection campaigns are performed at fixed intervals $\Delta t$.
2. $n_{I,C}$ hotspots are inspected during each inspection campaign.
3. Hotspots are prioritized for inspection according to a metric proposed by Bismut, Luque and Straub (2017), which is a function of a parameter $\eta$ as well as the structural importance and fatigue reliability of each hotspot.
4. An additional inspection campaign is launched if the predicted annual system failure probability exceeds a threshold $p_{th}$.
5. A maintenance campaign is launched if fatigue cracks are indicated and measured to be deeper than $a_R$.

Note that rules 4 and 5 have the following implication: Inspection information obtained at one hotspot contains indirect information on the fatigue state of the remaining hotspots as their fatigue behavior is correlated due to common influencing factors (see Straub and Faber, 2004 for a detailed discussion). Consider now the case in which a fatigue crack is unexpectedly indicated at one hotspot and measured to be deeper than $a_R$. Conditional on this inspection information, the probability that fatigue deterioration of the remaining hotspots has progressed faster than expected increases. Consequently, the system failure probability also increases. If it exceeds the threshold $p_{th}$, additional inspections and possibly repairs are performed as prescribed by rules 4 and 5. Because of these two rules and the explicit modeling of the dependence among the fatigue behavior of different hotspots, the current optimization of I&M of the frame captures scenarios in which the inspection and repair effort has to be increased due to accelerated fatigue deterioration.

From the above list of parameterized rules, it follows that the I&M strategy $\mathcal{S}$ is here fully defined by the parameters $\boldsymbol{\theta} = [\Delta t, p_{th}, n_{I,C}, \eta, a_R]^T$. To highlight the dependence of $\mathcal{S}$ on $\boldsymbol{\theta}$, we write $\mathcal{S}_{\boldsymbol{\theta}}$ in the following.

The utility function – or more precisely the cost function – underlying the current optimization is (generically) defined as:

$$C_2(\mathcal{S}_{\boldsymbol{\theta}}, \mathbf{X}, \mathbf{Y}, \mathbf{Z}, \mathbf{W}) = C_I(\mathcal{S}_{\boldsymbol{\theta}}, \mathbf{X}, \mathbf{Y}, \mathbf{Z}, \mathbf{W}) + C_M(\mathcal{S}_{\boldsymbol{\theta}}, \mathbf{X}, \mathbf{Y}, \mathbf{Z}, \mathbf{W}) + C_F(\mathcal{S}_{\boldsymbol{\theta}}, \mathbf{X}, \mathbf{Y}, \mathbf{Z}, \mathbf{W}) \qquad (16)$$

where $C_2(\mathcal{S}_{\boldsymbol{\theta}}, \mathbf{X}, \mathbf{Y}, \mathbf{Z}, \mathbf{W})$ is the total lifetime cost, $C_I(\mathcal{S}_{\boldsymbol{\theta}}, \mathbf{X}, \mathbf{Y}, \mathbf{Z}, \mathbf{W})$ is the total lifetime inspection cost, $C_M(\mathcal{S}_{\boldsymbol{\theta}}, \mathbf{X}, \mathbf{Y}, \mathbf{Z}, \mathbf{W})$ is the total lifetime maintenance cost and $C_F(\mathcal{S}_{\boldsymbol{\theta}}, \mathbf{X}, \mathbf{Y}, \mathbf{Z}, \mathbf{W})$ is the total lifetime cost of structural failure. These costs are defined in function of the I&M strategy $\mathcal{S}_{\boldsymbol{\theta}}$, the uncertain parameters $\mathbf{X}$ influencing the time-dependent system failure probability, the uncertain parameters $\mathbf{Y}$ influencing the effect of repairs and the uncertain parameters of the I&M cost model $\mathbf{W} = [\mathbf{W}_1^T, ..., \mathbf{W}_m^T]^T$, where $\mathbf{W}_j$ are the uncertain parameters influencing the I&M costs in year $j$ as defined in Table 5. The total I&M costs occurring in each year $j$ are evaluated based on the parametric cost models described in Section 4. It is here assumed that the different $\mathbf{W}_j$, $j = 1, ..., m$ are independent and identically distributed. Note that this assumption is not a limitation, as the model can be extended to account for dependent and non-identically distributed cost model parameters.

The optimal strategy $\mathcal{S}_{\theta^*}$ characterized by the optimal heuristic parameters $\boldsymbol{\theta}^*$ minimizes the expected value of the total lifetime cost $\mathbb{E}[C_2|\mathcal{S}_{\boldsymbol{\theta}}] = \mathbb{E}_{\mathbf{X},\mathbf{Y},\mathbf{Z},\mathbf{W}}[c_2(\mathcal{S}_{\boldsymbol{\theta}},\mathbf{X},\mathbf{Y},\mathbf{Z},\mathbf{W})]$. It follows that the optimal heuristic parameters $\boldsymbol{\theta}^*$ can be determined as:

$$\boldsymbol{\theta}^* = \arg\min_{\boldsymbol{\theta}} \mathbb{E}[C_2|\mathcal{S}_{\boldsymbol{\theta}}] \tag{17}$$

$\mathbb{E}[C_2|\mathcal{S}_{\boldsymbol{\theta}}]$ is evaluated as described in Appendix B.1. The optimal heuristic parameters $\boldsymbol{\theta}^*$ are here identified by conducting an exhaustive search across the following sets of parameter values: $\Delta t \in \{4,8\}$ [years], $p_{th} \in \{5\cdot 10^{-4}, 10^{-3}\}$, $n_{I,C} \in \{1,\dots,22\}$, $\eta = 1$ and $a_R = 1$ [mm]. The estimated expected total lifetime cost $\mathbb{E}[C_2|\mathcal{S}_{\boldsymbol{\theta}}]$ in function of $\boldsymbol{\theta}$ is shown in Figure 9.

All strategies with $n_{I,C} = \{3,4,5,6\}$ result in a similar expected total lifetime cost. This provides some flexibility to the decision-maker to choose a strategy based on their specific requirements regarding the inspection interval and structural reliability. Notably, both strategies with $\Delta t = 4$ years, exhibit similar expected costs for $n_{I,C} > 7$ regardless of the reliability threshold. When considering strategies with $\Delta t = 8$, the reliability threshold has an impact on the expected total lifetime cost: a lower threshold results in more unscheduled inspections between regular inspections. In our current example the optimal strategy $\mathcal{S}_{\theta^*}$ is characterized by $\boldsymbol{\theta}^* = [\Delta t = 8, p_{th} = 1\cdot 10^{-3}, n_{I,C} = 6, \eta = 1, a_R = 1]^T$.

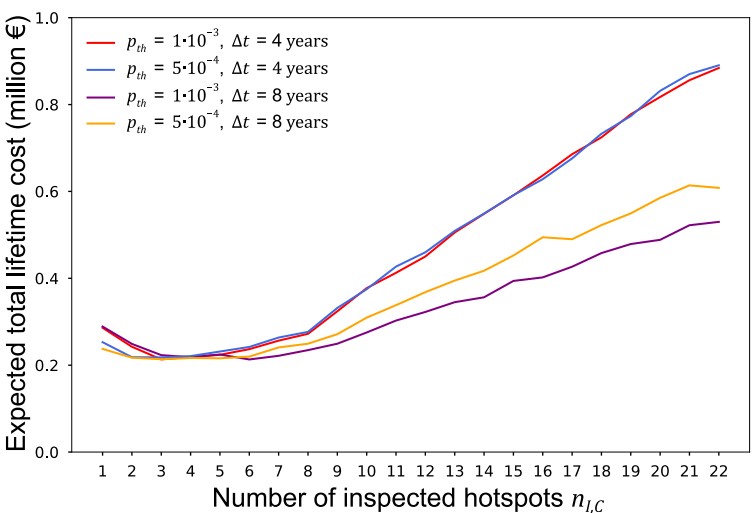

**Figure 9:** *Expected total lifetime cost $\mathbb{E}[C_2|\mathcal{S}_{\boldsymbol{\theta}}]$ in function of $\boldsymbol{\theta} = [\Delta t, p_{th}, n_{I,C}, \eta, a_R]^T$ determined based on the probabilistic I&M cost model*

Figure 10 shows the decomposed expected total lifetime cost $\mathbb{E}[C_2|\mathcal{S}_{\boldsymbol{\theta}}]$ in function of the heuristic parameters $\boldsymbol{\theta} = [\Delta t = 8, p_{th} = 1\cdot 10^{-3}, n_{I,C} = 1,\dots,22, \eta = 1, a_R = 1]^T$. This cost is composed of the expected values of the failure cost, inspection campaign cost, inspection operation cost, repair campaign cost, repair operation cost, and engineering cost for repairs. As the inspection effort increases (i.e., more hotspots are inspected during each inspection campaign), the expected value of the system failure cost (i.e., the risk of structural failure) decreases, and – as expected – the expected values of the inspection and repair costs increase. This nicely illustrates the impact of the risk mitigation measures on the structural risk of failure. Note that the engineering cost for repairs is constant in this case study since it is here incurred only once at the beginning of the operational phase as repair solutions are engineered proactively before the frame is commissioned. Consequently, they could be neglected in the current optimization, as they only shift the expected total lifetime costs upwards by a fixed value.

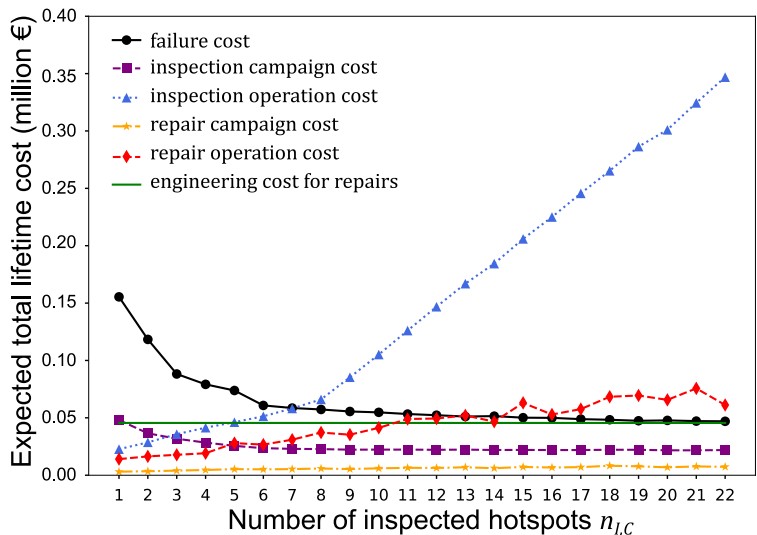

***Figure 10:*** *Decomposed expected total lifetime cost $\mathbb{E}[C_2|\mathcal{S}_\theta]$ in function of $\boldsymbol{\theta} = [\Delta t = 8, p_{th} = 1 \cdot 10^{-3}, n_{I,C} = 1, \dots, 22, \eta = 1, a_R = 1]^T$*

To support the decision on whether one should implement a I&M strategy, the predicted value of information and actions is computed by the difference between the expected total lifetime cost $\mathbb{E}[C_0]$ and the expected total lifetime cost $\mathbb{E}[C_2|\mathcal{S}_\theta]$. By normalizing this difference with respect to $\mathbb{E}[C_0]$, the relative $\bar{V}_{SS-A}^{PIPA-DA}(\boldsymbol{\theta})$ is obtained (Farhan, Schneider and Thöns, 2021):

$$\bar{V}_{SS-A}^{PIPA-DA}(\boldsymbol{\theta}) = \frac{\mathbb{E}[C_0] - \mathbb{E}[C_2|\mathcal{S}_\theta]}{\mathbb{E}[C_0]} \tag{18}$$

Figure 11 shows the $\bar{V}_{SS-A}^{PIPA-DA}$ in function of the parameters $\boldsymbol{\theta} = [\Delta t = 8, p_{th} = 1 \cdot 10^{-3}, n_{I,C} = 1, \dots, 22, \eta = 1, a_R = 1]^T$, where $\Delta t = 8$ and $p_{th} = 1 \cdot 10^{-3}$ are the optimal inspection interval and reliability threshold. The dashed blue line corresponds to the expected total lifetime cost $\mathbb{E}[C_0]$ determined by the SS-A. The dashed-dotted blue line corresponds to the expected total lifetime cost $\mathbb{E}[C_2|\mathcal{S}_\theta]$. Notably, $\bar{V}_{SS-A}^{PIPA-DA}$ is positive for $n_{I,C} = 1, \dots, 22$. This result indicates that it is a rational decision to inspect and maintain the frame. As expected, the highest $\bar{V}_{SS-A}^{PIPA-DA}$ is obtained when implementing the optimal strategy $\mathcal{S}_{\theta^*}$ with an optimal number of inspected hotspots in each inspection campaign $n_{I,C} = 6$.

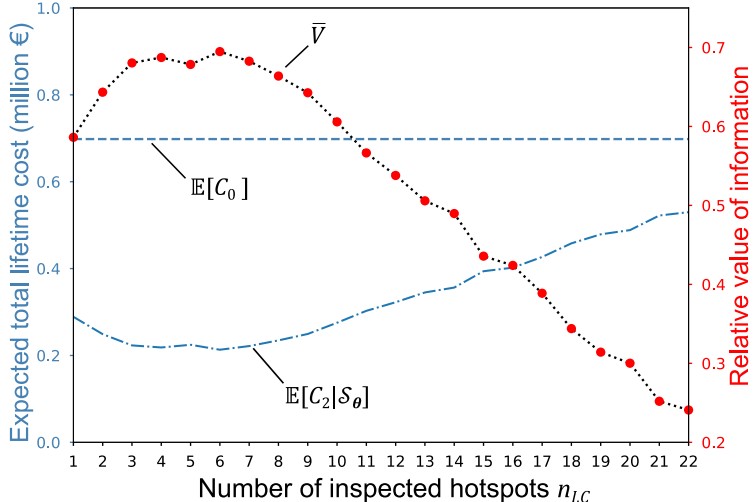

**Figure 11:** *Relative DV $\bar{V}_{SS-A}^{PIPA-DA}(\boldsymbol{\theta}) = (\mathbb{E}[C_0] - \mathbb{E}[C_2|\mathcal{S}_{\boldsymbol{\theta}}])/\mathbb{E}[C_0]$ together with the expected lifetime costs $\mathbb{E}[C_2|\mathcal{S}_{\boldsymbol{\theta}}]$ in function of $\boldsymbol{\theta} = [\Delta t = 8, p_{th} = 1 \cdot 10^{-3}, n_{I,C} = 1, \dots, 22, \eta = 1, a_R = 1]^T$ and $E[C_0]$*

## 6.3 Predictive information and predictive action decision analysis considering expected I&M costs

The numerical example considers only a single turbine support structure. In this case, the cost models defined in Eq. (11) and (12) can be expressed as linear functions of the number of inspected and repaired hotspots as follows:

$$C_I = C_C + n_{I,C} \cdot C_{I,Op} \quad \text{with} \quad C_{I,Op} = \frac{t_{I,C} \cdot (1 + WD) \cdot C_{\text{shift}}}{t_{\text{shift}}} \tag{19}$$

and

$$C_M = C_C + C_E + n_{M,C} \cdot C_{M,Op} \quad \text{with} \quad C_{M,Op} = \frac{t_{M,C} \cdot (1 + WD) \cdot C_{\text{shift}}}{t_{\text{shift}}} \tag{20}$$

The expected lifetime cost $\mathbb{E}[C_2|\mathcal{S}_{\boldsymbol{\theta}}]$ can now be estimated based on the expected values of the parameters of the I&M cost model as described in Appendix B.2. Subsequently, the optimal heuristic parameters $\boldsymbol{\theta}^*$ are identified based on Eq. (17) by conducting again an exhaustive search across the following sets of parameter values: $\Delta t \in \{4, 8\}$ [years], $p_{th} \in \{5 \cdot 10^{-4}, 10^{-3}\}, n_{I,C} \in \{1, \dots, 22\}, \eta = 1$ and $a_R = 1$ [mm]. The estimated expected total lifetime cost $\mathbb{E}[C_2|\mathcal{S}_{\boldsymbol{\theta}}]$ considering expected I&M costs are shown in Figure 12.

It can be seen that the current analysis provides the same results as the analysis considering the probabilistic I&M costs (cf. also Figure 9) and thus the same optimal strategy $\mathcal{S}_{\boldsymbol{\theta}^*}$ with $\boldsymbol{\theta}^* = [\Delta t = 8, p_{th} = 1 \cdot 10^{-3}, n_{I,C} = 6, \eta = 1, a_R = 1]^T$. Consequently, the current analysis illustrates that the I&M costs can be considered deterministically as expected values in the DV analysis if they are included in the optimization on a linear basis.

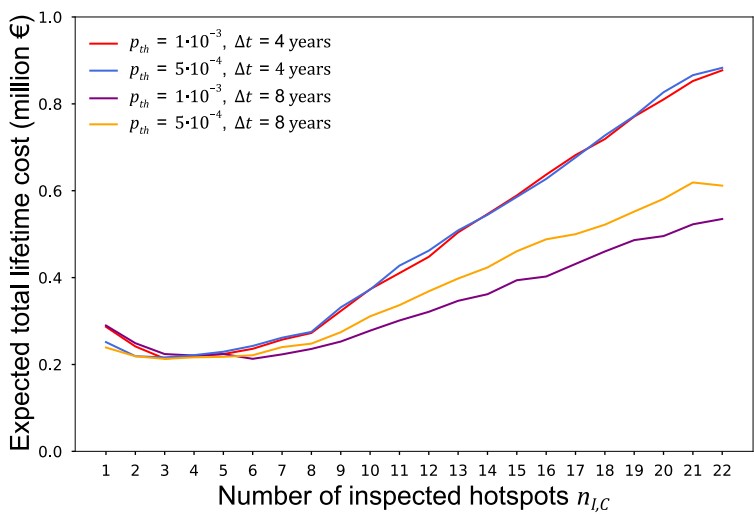

***Figure 12:*** *Expected total lifetime cost* $\mathbb{E}[C_2|\mathcal{S}_\theta]$ *in function of* $\boldsymbol{\theta} = [\Delta t, p_{th}, n_I, \eta, a_R]^T$ *determined based on the expected values of the parameters of the I&M cost model*

Aligning with existing works, we utilize the probabilistic I&M cost model to derive deterministic or normalized cost ratios. In the literature, such a normalization is typically performed with respect to the failure cost or expected campaign cost due to their significant contribution to the overall lifetime costs. Applying the same methodology, we obtain the normalized cost models summarized in Table 7 and Table 8. These models can subsequently be used to optimize I&M of offshore wind turbine support structures if the costs are included in the underlying models on a linear basis.

***Table 7.*** *Cost model normalized with respect to the expected campaign cost*

| Cost parameter | Ratio | Normalized value |
|---|---|---|
| Campaign cost $c_C$ | $9.11 \cdot 10^3 / 9.11 \cdot 10^3$ | 1.00 |
| Failure cost $c_F$ | $2.00 \cdot 10^7 / 9.11 \cdot 10^3$ | 2193.77 |
| Engineering cost $c_E$ | $4.55 \cdot 10^4 / 9.11 \cdot 10^3$ | 5.00 |
| Inspection cost below water with EM $c_{I,Op}$ | $8.87 \cdot 10^3 / 9.11 \cdot 10^3$ | 0.97 |
| Inspection cost above water with EM $c_{I,Op}$ | $3.54 \cdot 10^3 / 9.11 \cdot 10^3$ | 0.38 |
| Repair cost below water with welding $c_{M,Op}$ | $4.58 \cdot 10^4 / 9.11 \cdot 10^3$ | 5.02 |
| Repair cost above water with welding $c_{M,Op}$ | $3.80 \cdot 10^4 / 9.11 \cdot 10^3$ | 4.17 |

***Table 8.*** *Cost model normalized with respect to the failure cost*

| Cost parameter | Ratio | Normalized value |
|---|---|---|
| Failure cost $c_F$ | $2.00 \cdot 10^7 / 2.00 \cdot 10^7$ | 1.0 |
| Campaign cost $c_C$ | $9.11 \cdot 10^3 / 2.00 \cdot 10^7$ | $4.55 \cdot 10^{-4}$ |
| Engineering cost $c_E$ | $4.55 \cdot 10^4 / 2.00 \cdot 10^7$ | $2.28 \cdot 10^{-3}$ |

| | | |
|---|---|---|
| Inspection cost below water with EM $c_{I,Op}$ | $8.87 \cdot 10^3 / 2.00 \cdot 10^7$ | $4.43 \cdot 10^{-4}$ |
| Inspection cost above water with EM $c_{I,Op}$ | $3.54 \cdot 10^3 / 2.00 \cdot 10^7$ | $1.77 \cdot 10^{-4}$ |
| Repair cost below water with welding $c_{M,Op}$ | $4.58 \cdot 10^4 / 2.00 \cdot 10^7$ | $2.29 \cdot 10^{-3}$ |
| Repair cost above water with welding $c_{M,Op}$ | $3.80 \cdot 10^4 / 2.00 \cdot 10^7$ | $1.90 \cdot 10^{-3}$ |

# 7 Summary and concluding remarks

This paper formulates and applies a probabilistic cost model to support the planning of I&M of the turbine support structures in offshore wind farms. It provides a decision-theoretical basis for optimizing I&M activities, with an emphasis on integrating the probabilistic cost model in the decision analysis. The probabilistic cost model is derived based on a discussion of (a) the types of I&M of turbine support structures and (b) the parameters that influence the overall I&M cost. Subsequently, variance-based sensitivity analyses are performed based on the probabilistic cost model to quantify the influence of the different cost model parameters on the overall I&M costs. Finally, the proposed probabilistic cost model is applied in a numerical example in which the I&M regime is optimized for a frame with steel members which resembles a jacket support structure of an offshore wind turbine. As part of the example, a SS-A, PIPA-DA and DV analysis is performed. The SS-A determines the lifetime risk of structural when no information is collected, and no maintenance actions are performed throughout the structure's lifetime. The PIPA-DA optimizes a heuristic I&M strategy defined by parameterized rules that guide the actions to be taken based on the available system information. The analysis is first performed based on the probabilistic model of the I&M costs. Subsequently, it is performed based on the expected values of the I&M costs. This is here possible since the costs are included in the model on a linear basis. Both analyses yield the same optimal heuristic I&M strategy. Finally, to determine the cost-effectiveness of the identified optimal I&M strategy, a DV analysis is carried out considering the probabilistic I&M cost model.

Based on our work, the following conclusions can be drawn:

1. The generic framework described in Section 2 facilitates an optimization of I&M regimes for turbine support structures in offshore wind farms taking into account the uncertainties in the I&M costs.

2. The proposed probabilistic cost model can be utilized to quantify I&M costs at wind farm, structural system, and component level, which can be updated when the new data on the parameters governing the I&M costs become available during the operation of wind farms.

3. The sensitivity analyses showed that, at component level, the campaign cost and engineering cost have the highest influence on the overall I&M cost, while the vessel cost per shift has the highest impact on the overall I&M costs at structural system and wind farm level.

4. The decision analysis described in the numerical example identifies a cost and risk optimal I&M strategy for a steel frame subject to fatigue based on the probabilistic cost model. An optimal inspection interval of $\Delta t = 8$ years is obtained from the PIPA-DA and DV analysis. Furthermore, if the annual system failure probability exceeds a threshold of $p_{th} = 1 \cdot 10^{-3}$ yr$^{-1}$, an additional inspection campaign is launched. In each

campaign, six prioritized hotspots are inspected, and a repair campaign is launched if fatigue cracks are indicated and measured to be deeper than $a_R = 1$ mm.

5. With the help of the numerical example, it is demonstrated that the I&M costs can be considered deterministically as expected values in the decision analysis if they are included in the optimization on a linear basis.

6. The expected I&M costs at the structural system level depend solely on the number of campaigns and components involved in the I&M operations as wells as on the expected campaign, engineering, and operational cost, which therefore can be normalized and used in decision analyses to optimize the I&M regime of support structures at the structural system level. In the future, we will research similar concepts to derive deterministic (normalized) cost models for I&M planning at wind farm level.

## Author contribution

**Muhammad Farhan:** Conceptualization, Methodology, Software, Formal analysis, Data curation, Validation, Writing - original draft. **Ronald Schneider:** Conceptualization, Methodology, Software, Formal analysis, Data curation, Validation, Supervision, Writing – review & editing. **Sebastian Thöns:** Funding acquisition, Supervision, Writing – review & editing. **Max Gündel:** Supervision, Writing – review & editing.

## Competing interests

The contact author has declared that none of the authors has any competing interests.

## Acknowledgements

This work was supported by the German Ministry for Economic Affairs and Climate Actions (BMWK) through grant 03SX449Z and Projektträger Jülich (PtJ). We would like to thank the BMWK and PtJ as well as our project partners Cubert GmbH, Fraunhofer IOSB, Wölfel Engineering GmbH & Co. KG, Baltic Taucherei- und Bergungsbetrieb Rostock GmbH, SLV Mecklenburg-Vorpommern GmbH and RWE Renewables GmbH for their support and collaboration in the project.

## Appendix A        Expected utilities and optimal I&M decisions

As described in Section 2.1, the expected utilities $\mathbb{E}[U_0]$, $\mathbb{E}[U_1|\mathbf{a}]$ and $\mathbb{E}[U_2|i]$ form the basis for utility-informed optimizations of inspection/monitoring and maintenance of WT support structures in offshore wind farms. This appendix briefly summarizes the evaluation of these expected utilities. The summary assumes that the corresponding utility functions and the probability distributions of their input parameters are available. In addition, the appendix shows in more detail how decisions on inspection/monitoring and maintenance are optimized by maximizing the corresponding expected utilities.

### A.1   System state analysis

As part of the system state analysis (SS-A), the expected utility $\mathbb{E}[U_0]$ is evaluated as follows:

$$\mathbb{E}[U_0] = \mathbb{E}_{\mathbf{X},\mathbf{W}}[U_0(\mathbf{X},\mathbf{W})] = \int_{\mathbf{X}} \int_{\mathbf{W}} U_0(\mathbf{x},\mathbf{w})\, p(\mathbf{x},\mathbf{w})\, \mathrm{d}\mathbf{w}\, \mathrm{d}\mathbf{x} \tag{21}$$

where $\mathbb{E}_{\mathbf{X},\mathbf{W}}[U_0(\mathbf{X},\mathbf{W})]$ is the expected value of the utility function $U_0(\mathbf{X},\mathbf{W})$ defined in Eq. (1) with respect random variables $\mathbf{X}$ and $\mathbf{W}$ which influence the system state and the I&M costs, and $p(\mathbf{x},\mathbf{w})$ is the joint prior probability distribution of $\mathbf{X}$ and $\mathbf{W}$.

## A.2 Predicted action decision analysis

In the predicted action decision analysis (PA-DA), the expected utility $\mathbb{E}[U_1|\mathbf{a}]$ is evaluated. This expected utility is conditional on a possible choice of maintenance actions $\mathbf{a}$, and computed with respect to the random variables $\mathbf{Y}$, $\mathbf{X}$ and $\mathbf{W}$. These random variables influence the outcome of the associated maintenance actions, the system state, and the I&M costs. In a generic format, $\mathbb{E}[U_1|\mathbf{a}]$ is determined as:

$$\mathbb{E}[U_1|\mathbf{a}] = \mathbb{E}_{\mathbf{Y},\mathbf{X},\mathbf{W}}[U_1(\mathbf{a},\mathbf{Y},\mathbf{X},\mathbf{W})] = \int_{\mathbf{Y}} \int_{\mathbf{X}} \int_{\mathbf{W}} U_1(\mathbf{a},\mathbf{y},\mathbf{x},\mathbf{w})\, p(\mathbf{y},\mathbf{x},\mathbf{w})\, \mathrm{d}\mathbf{w}\, \mathrm{d}\mathbf{x}\, \mathrm{d}\mathbf{y} \qquad (22)$$

where $\mathbb{E}_{\mathbf{Y},\mathbf{X},\mathbf{W}}[U_1(\mathbf{a},\mathbf{Y},\mathbf{X},\mathbf{W})]$ is the expected value of the utility function $U_1(\mathbf{a},\mathbf{Y},\mathbf{X},\mathbf{W})$ defined in Eq. (2) with respect to $\mathbf{Y}$, $\mathbf{X}$, and $\mathbf{W}$, and $p(\mathbf{y},\mathbf{x},\mathbf{w})$ is the joint prior probability distribution of $\mathbf{Y}$, $\mathbf{X}$ and $\mathbf{W}$.

The optimal maintenance actions $\mathbf{a}^*$ are identified by maximizing the conditional expected value of $U_1$ as:

$$\mathbf{a}^* = \arg\max_{\mathbf{a}} \mathbb{E}[U_1|\mathbf{a}] \qquad (23)$$

Finally, the expected value of $U_1$ conditional on the optimal maintenance actions $\mathbf{a}^*$ can be defined as:

$$\mathbb{E}[U_1|\mathbf{a}^*] = \mathbb{E}_{\mathbf{Y},\mathbf{X},\mathbf{W}}[U_1(\mathbf{a}^*,\mathbf{Y},\mathbf{X},\mathbf{W})] \qquad (24)$$

## A.3 Predicted information and predicted action decision analysis

A predicted information and predicted action decision analysis (PIPA-DA) is performed to jointly optimize decisions on inspection/monitoring and maintenance (see also Thöns and Kapoor, 2019). In this analysis, the expected value of the utility $U_2$ is maximized based on predicted inspection/monitoring outcomes and predicted maintenance actions. When applying the extensive form of the analysis based on the lower branch of the decision tree in Figure 1 (Raiffa and Schlaifer, 1961), the optimization is progressed from the leaf of the branch towards the node representing the decision on the inspection/monitoring regime $i$. The analysis starts by determining the expected value of $U_2$ conditional on a choice of the inspection/monitoring regime $i$, a realization of the corresponding inspection/monitoring outcomes $\mathbf{Z}_i = \mathbf{z}_i$, and a possible choice of maintenance actions $\mathbf{a}$ as follows:

$$\mathbb{E}[U_2|i,\mathbf{z}_i,\mathbf{a}] = \mathbb{E}_{\mathbf{Y},\mathbf{X},\mathbf{W}|\mathbf{Z}_i=\mathbf{z}_i}[U_2(i,\mathbf{z}_i,\mathbf{a},\mathbf{Y},\mathbf{X},\mathbf{W})]$$
$$= \int_{\mathbf{Y}} \int_{\mathbf{X}} \int_{\mathbf{W}} U_2(i,\mathbf{z}_i,\mathbf{a},\mathbf{Y},\mathbf{X},\mathbf{W})\, p(\mathbf{y},\mathbf{x},\mathbf{w}|\mathbf{z}_i)\, \mathrm{d}\mathbf{w}\, \mathrm{d}\mathbf{x}\, \mathrm{d}\mathbf{y} \qquad (25)$$

where $\mathbb{E}_{\mathbf{Y},\mathbf{X},\mathbf{W}|\mathbf{Z}_i=\mathbf{z}_i}[U_2(i,\mathbf{z}_i,\mathbf{a},\mathbf{Y},\mathbf{X},\mathbf{W})]$ is the conditional expected value of the utility function $U_2(i,\mathbf{z}_i,\mathbf{a},\mathbf{Y},\mathbf{X},\mathbf{W})$ defined in Eq. (3) with respect to $\mathbf{Y}$, $\mathbf{X}$ and $\mathbf{W}$ conditional on $\mathbf{Z}_i = \mathbf{z}_i$, and $p(\mathbf{y},\mathbf{x},\mathbf{w}|\mathbf{z}_i)$ is the joint posterior probability distribution of $\mathbf{Y}$, $\mathbf{X}$ and $\mathbf{W}$ given $\mathbf{Z}_i = \mathbf{z}_i$, which is determined using Bayesian analysis.

Subsequently, the optimal maintenance actions $\mathbf{a}^*_{|i,\mathbf{z}_i}$ conditional on a certain choice of the inspection/monitoring regime $i$ and a corresponding realization of the inspection/monitoring outcomes $\mathbf{Z}_i = \mathbf{z}_i$ can be determined by maximizing $\mathbb{E}[U_2|i,\mathbf{z}_i,\mathbf{a}]$ as:

$$\mathbf{a}^*_{|i,\mathbf{z}_i} = \arg\max_{\mathbf{a}} \mathbb{E}[U_2|i,\mathbf{z}_i,\mathbf{a}] \tag{26}$$

Given that the decision-maker upon knowing the inspection/monitoring outcomes $\mathbf{Z}_i = \mathbf{z}_i$ will always make the optimal maintenance decisions $\mathbf{a}^*_{|i,\mathbf{z}_i}$, the expected value of the utility $U_2$ conditional on a choice of the inspection/monitoring regime $i$ is computed as:

$$\mathbb{E}[U_2|i] = \mathbb{E}_{\mathbf{Z}_i}[\mathbb{E}_{\mathbf{Y},\mathbf{X},\mathbf{W}|\mathbf{Z}_i}[U_2(i,\mathbf{Z}_i,\mathbf{a}^*_{|i,\mathbf{Z}_i},\mathbf{Y},\mathbf{X},\mathbf{W})]]$$

$$= \int_{\mathbf{Z}_i} \left[ \max_{\mathbf{a}} \mathbb{E}_{\mathbf{Y},\mathbf{X},\mathbf{W}|\mathbf{Z}_i=\mathbf{z}_i}[U_2(i,\mathbf{z}_i,\mathbf{a},\mathbf{Y},\mathbf{X},\mathbf{W})] \right] p(\mathbf{z}_i)\,\mathrm{d}\mathbf{z}_i \tag{27}$$

where $\mathbb{E}_{\mathbf{Z}_i}[\cdot]$ is the expectation with respect to $\mathbf{Z}_i$, and $p(\mathbf{z}_i)$ is the marginal probability distribution of the probabilistic inspection/monitoring outcomes $\mathbf{Z}_i$. The optimal inspection/maintenance regime $i^*$ is then obtained by maximizing $\mathbb{E}[U_2|i]$ as:

$$i^* = \arg\max_i \mathbb{E}[U_2|i] \tag{28}$$

Finally, the maximum expected value of $U_2$ conditional on $i^*$ is obtained as:

$$\mathbb{E}[U_2|i^*] = \mathbb{E}_{\mathbf{Z}_{i^*}}[\mathbb{E}_{\mathbf{Y},\mathbf{X},\mathbf{W}|\mathbf{Z}_{i^*}}[U_2(i^*,\mathbf{Z}_{i^*},\mathbf{a}^*_{|i^*,\mathbf{Z}_{i^*}},\mathbf{Y},\mathbf{X},\mathbf{W})]] \tag{29}$$

From Eq. (26), (27) and (28) it can be seen that a PIPA-DA cannot be summarized in a single optimization problem if the extensive form of the analysis is applied. As shown in Eq. (26), the optimal decisions on the maintenance actions $\mathbf{a}^*_{|i,\mathbf{z}_i}$ can in this case only be determined conditional on a certain realization of the inspection/monitoring outcomes $\mathbf{Z}_i = \mathbf{z}_i$.

# Appendix B    Expected total lifetime cost conditional on a heuristic I&M strategy

The expected value of the total lifetime cost $\mathbb{E}[C_2|\mathcal{S}_{\boldsymbol{\theta}}]$ conditional on a heuristic I&M strategy $\mathcal{S}_{\boldsymbol{\theta}}$ is required to identify the cost and risk-informed optimal heuristic I&M strategy $\mathcal{S}_{\boldsymbol{\theta}^*}$ as described in numerical example in Section 6. In the following, we outline the evaluation of $\mathbb{E}[C_2|\mathcal{S}_{\boldsymbol{\theta}}]$ considering probabilistic and expected I&M costs.

## B.1  Probabilistic I&M costs

The expected total lifetime cost $\mathbb{E}[C_2|\mathcal{S}_{\boldsymbol{\theta}}]$ considering probabilistic I&M costs is defined as:

$$\mathbb{E}[C_2|\mathcal{S}_{\boldsymbol{\theta}}] = \mathbb{E}_{\mathbf{X},\mathbf{Y},\mathbf{Z},\mathbf{W}}[C_2(\mathcal{S}_{\boldsymbol{\theta}},\mathbf{X},\mathbf{Y},\mathbf{Z},\mathbf{W})]$$

$$= \int_{\mathbf{X}}\int_{\mathbf{Y}}\int_{\mathbf{Z}}\int_{\mathbf{W}} C_2(\mathcal{S}_{\boldsymbol{\theta}},\mathbf{X},\mathbf{Y},\mathbf{Z},\mathbf{W})\,p(\mathbf{x},\mathbf{y},\mathbf{z}|\mathcal{S}_{\boldsymbol{\theta}})\,p(\mathbf{w}|\mathcal{S}_{\boldsymbol{\theta}})\,\mathrm{d}\mathbf{x}\,\mathrm{d}\mathbf{y}\,\mathrm{d}\mathbf{z}\,\mathrm{d}\mathbf{w} \tag{30}$$

in which $p(\mathbf{x},\mathbf{y},\mathbf{z}|\mathcal{S}_{\boldsymbol{\theta}})\,p(\mathbf{w}|\mathcal{S}_{\boldsymbol{\theta}})$ is the joint probability density function (PDF) of $\mathbf{X}$, $\mathbf{Y}$, $\mathbf{Z}$ and $\mathbf{W}$. Eq. (30) implies that the uncertain parameters $\mathbf{W}$ governing the I&M costs are modeled as statistically independent of the uncertain parameters $\mathbf{X}$ influencing the system failure

probability, the uncertain parameters $\mathbf{Y}$ affecting the repair outcomes and the probabilistic inspection outcomes $\mathbf{Z}$.

Eq. (30) can be rewritten as:

$$\mathbb{E}[C_2|\mathcal{S}_\theta] = \int_{\mathbf{Z}} \int_{\mathbf{W}} \mathbb{E}[C_2|\mathcal{S}_\theta, \mathbf{z}, \mathbf{w}] \; p(\mathbf{z}|\mathcal{S}_\theta) \, p(\mathbf{w}|\mathcal{S}_\theta) \, \mathrm{d}\mathbf{z} \, \mathrm{d}\mathbf{w} \qquad (31)$$

where $\mathbb{E}[C_2|\mathcal{S}_\theta, \mathbf{z}, \mathbf{w}]$ is the expected total lifetime cost conditional inspection outcomes $\mathbf{Z} = \mathbf{z}$ and corresponding repairs as prescribed by strategy $\mathcal{S}_\theta$ and $p(\mathbf{z}|\mathcal{S}_\theta)$ is the marginal PDF of the lifetime inspection outcomes.

$\mathbb{E}[C_2|\mathcal{S}_\theta, \mathbf{z}, \mathbf{w}]$ is computed as:

$$\mathbb{E}[C_2|\mathcal{S}_\theta, \mathbf{z}, \mathbf{w}] = \mathbb{E}_{\mathbf{X},\mathbf{Y}|\mathbf{Z}=\mathbf{z},\, \mathbf{W}=\mathbf{w}}[C_2(\mathcal{S}_\theta, \mathbf{X}, \mathbf{Y}, \mathbf{z}, \mathbf{w})] = \int_{\mathbf{X}} \int_{\mathbf{Y}} C_2(\mathcal{S}_\theta, \mathbf{X}, \mathbf{Y}, \mathbf{z}, \mathbf{w}) \; p(\mathbf{x}, \mathbf{y}|\mathcal{S}_\theta, \mathbf{z}) \, \mathrm{d}\mathbf{x} \, \mathrm{d}\mathbf{y} \quad (32)$$

where $p(\mathbf{x}, \mathbf{y}|\mathcal{S}_\theta, \mathbf{z})$ is the conditional PDF of $\mathbf{X}$ and $\mathbf{Y}$ given $\mathbf{Z} = \mathbf{z}$. $\mathbb{E}[C_2|\mathcal{S}_\theta, \mathbf{z}, \mathbf{w}]$ can be decomposed as:

$$\mathbb{E}[C_2|\mathcal{S}_\theta, \mathbf{z}, \mathbf{w}] = \mathbb{E}[C_I|\mathcal{S}_\theta, \mathbf{z}, \mathbf{w}] + \mathbb{E}[C_M|\mathcal{S}_\theta, \mathbf{z}, \mathbf{w}] + \mathbb{E}[C_F|\mathcal{S}_\theta, \mathbf{z}] \qquad (33)$$

where $\mathbb{E}[C_I|\mathcal{S}_\theta, \mathbf{z}, \mathbf{w}]$ is the conditional expected lifetime inspection cost, $\mathbb{E}[C_M|\mathcal{S}_\theta, \mathbf{z}, \mathbf{w}]$ is the conditional expected lifetime maintenance cost, and $\mathbb{E}[C_F|\mathcal{S}_\theta, \mathbf{z}]$ quantifies the conditional expected lifetime failure costs over the lifetime of the structure. Note that the latter does not depend on the I&M cost model parameters as shown in Eq. (36) below.

The conditional expected lifetime inspection cost $\mathbb{E}[C_I|\mathcal{S}, \mathbf{z}, \mathbf{w}]$ is computed as:

$$\mathbb{E}[C_I|\mathcal{S}_\theta, \mathbf{z}, \mathbf{w}] = \sum_{j=1}^{m} C_{I,j}(\mathcal{S}_\theta, \mathbf{z}, \mathbf{w}) \cdot \gamma_j \cdot [1 - \Pr(F_j|\mathcal{S}_\theta, \mathbf{z})] \qquad (34)$$

where the $j$th term represents the inspection costs in year $j$ given that failure has not occurred up to the end of that year; $C_{I,j}(\mathcal{S}_\theta, \mathbf{z}, \mathbf{w})$ is the inspection cost in year $j$, which are estimated based on the model defined in Eq. (11); and $1 - \Pr(F_j|\mathcal{S}_\theta, \mathbf{z})$ is the probability of survival of the system up to the end of year $j$ conditional on the inspection outcomes $\mathbf{Z} = \mathbf{z}$ and corresponding repairs as determined by the strategy $\mathcal{S}$.

Similarly, the conditional expected lifetime maintenance cost $\mathbb{E}[C_M|\mathcal{S}_\theta, \mathbf{z}, \mathbf{w}]$ is given by:

$$\mathbb{E}[C_M|\mathcal{S}_\theta, \mathbf{z}, \mathbf{w}] = \sum_{j=1}^{m} C_{M,j}(\mathcal{S}_\theta, \mathbf{z}, \mathbf{w}) \cdot \gamma_j \cdot [1 - \Pr(F_j|\mathcal{S}_\theta, \mathbf{z})] \qquad (35)$$

where $C_{M,j}(\mathcal{S}_\theta, \mathbf{z}, \mathbf{w})$ is the maintenance costs in year $j$, which are determined based on the model defined in Eq. (12).

The conditional expected lifetime failure cost $\mathbb{E}[C_F|\mathcal{S}_\theta, \mathbf{z}, \mathbf{w}]$ is evaluated as:

$$\mathbb{E}[C_F|\mathcal{S}_{\boldsymbol{\theta}},\mathbf{z}] = \sum_{j=1}^{m} c_F \cdot \gamma_j \cdot [\Pr(F_j|\mathcal{S}_{\boldsymbol{\theta}},\mathbf{z}) - \Pr(F_{j-1}|\mathcal{S}_{\boldsymbol{\theta}},\mathbf{z})] \tag{36}$$

where $c_F$ is the deterministic failure cost and $\Pr(F_j|\mathcal{S}_{\boldsymbol{\theta}},\mathbf{z}) - \Pr(F_{j-1}|\mathcal{S}_{\boldsymbol{\theta}},\mathbf{z})$ is the probability of failure for year $j$ given $\mathbf{Z} = \mathbf{z}$ (cf. also Eq. (15)).

$\mathbb{E}[C_2|\mathcal{S}_{\boldsymbol{\theta}}]$ is here estimated using a MC approach:

$$\mathbb{E}[C_2|\mathcal{S}_{\boldsymbol{\theta}}] = \int_{\mathbf{Z}} \int_{\mathbf{W}} \mathbb{E}[C_2|\mathcal{S}_{\boldsymbol{\theta}},\mathbf{z},\mathbf{w}] \; p(\mathbf{z}|\mathcal{S}_{\boldsymbol{\theta}}) \, p(\mathbf{w}|\mathcal{S}_{\boldsymbol{\theta}}) \, \mathrm{d}\mathbf{z} \, \mathrm{d}\mathbf{w} \approx \frac{1}{n}\sum_{i=1}^{n} \mathbb{E}[C_2|\mathcal{S}_{\boldsymbol{\theta}},\mathbf{z}^{(i)},\mathbf{w}^{(i)}] \tag{37}$$

in which $\{\mathbf{z}^{(i)}\}_{i=1}^{n}$ are samples of the probabilistic inspection outcomes $\mathbf{Z}$ conditional on the heuristic strategy $\mathcal{S}_{\boldsymbol{\theta}}$, which are generated as discussed by Bismut and Straub (2021); $\{\mathbf{w}^{(i)}\}_{i=1}^{n}$ are samples of the uncertain cost model parameters $\mathbf{W} = [\mathbf{W}_1^T, \dots, \mathbf{W}_m^T]^T$, where $\mathbf{W}_j$ are the probabilistic parameters influencing the I&M costs in year $j$ as defined in Table 5. As described in Section 6.2, it is assumed that the different $\mathbf{W}_j, j = 1, \dots, m$ are independent and identically distributed. Thus, the joint PDF $p(\mathbf{w}|\mathcal{S}_{\boldsymbol{\theta}})$ can simply be written as $p(\mathbf{w}|\mathcal{S}_{\boldsymbol{\theta}}) = p(\mathbf{w}_1|\mathcal{S}_{\boldsymbol{\theta}}) \cdot \dots \cdot p(\mathbf{w}_m|\mathcal{S}_{\boldsymbol{\theta}})$. It is further assumed that repair solutions are pre-engineered before the frame is commissioned. Thus, the engineering costs are only incurred once at the beginning of the lifetime.

The expected lifetime cost $\mathbb{E}[C_2|\mathcal{S}_{\boldsymbol{\theta}}]$ is in this contribution estimated using MCS with 400 samples of the inspection outcome $\mathbf{Z}$ and cost model parameters $\mathbf{W}$.

## B.1 Expected I&M costs

Based on Eq. (19), the conditional expected lifetime inspection cost can be formulated such that it only depends on the inspection outcomes $\mathbf{Z} = \mathbf{z}$ and corresponding repairs as determined by the strategy $\mathcal{S}_{\boldsymbol{\theta}}$, i.e.:

$$\mathbb{E}[C_I|\mathcal{S}_{\boldsymbol{\theta}},\mathbf{z}] = \mathbb{E}_{\mathbf{W}|\mathbf{Z}=\mathbf{z}}[C_I|\mathcal{S}_{\boldsymbol{\theta}},\mathbf{z},\mathbf{W}] = \sum_{j=1}^{m} \mathbb{E}_{\mathbf{W}|\mathbf{Z}=\mathbf{z}}[C_{I,j}(\mathcal{S}_{\boldsymbol{\theta}},\mathbf{z},\mathbf{W})] \cdot \gamma_j \cdot [1 - \Pr(F_j|\mathcal{S}_{\boldsymbol{\theta}},\mathbf{z})] \tag{38}$$

with

$$\mathbb{E}_{\mathbf{W}|\mathbf{Z}=\mathbf{z}}[C_{I,j}(\mathcal{S}_{\boldsymbol{\theta}},\mathbf{z},\mathbf{W})] = n_{C,j}(\mathcal{S}_{\boldsymbol{\theta}},\mathbf{z}) \cdot \mathbb{E}[C_C] + n_{I,C,j}(\mathcal{S}_{\boldsymbol{\theta}},\mathbf{z}) \cdot \mathbb{E}[C_{I,Op}] \tag{39}$$

where $n_{C,j}(\mathcal{S}_{\boldsymbol{\theta}},\mathbf{z})$ and $n_{I,C,j}(\mathcal{S}_{\boldsymbol{\theta}},\mathbf{z})$ are the total numbers of inspection campaigns and component inspections in year $j$.

Equivalently, based on Eq. (20), the conditional expected lifetime maintenance cost can be expressed as:

$$\mathbb{E}[C_M|\mathcal{S}_{\boldsymbol{\theta}},\mathbf{z}] = \mathbb{E}_{\mathbf{W}|\mathbf{Z}=\mathbf{z}}[C_M|\mathcal{S}_{\boldsymbol{\theta}},\mathbf{z},\mathbf{W}] = \sum_{j=1}^{m} \mathbb{E}_{\mathbf{W}|\mathbf{Z}=\mathbf{z}}[C_{M,j}(\mathcal{S}_{\boldsymbol{\theta}},\mathbf{z},\mathbf{W})] \cdot \gamma_j \cdot [1 - \Pr(F_j|\mathcal{S}_{\boldsymbol{\theta}},\mathbf{z})] \tag{40}$$

with

$$\mathbb{E}_{\mathbf{W}|\mathbf{Z}=\mathbf{z}}[c_{M,j}(\mathcal{S}_{\boldsymbol{\theta}},\mathbf{z},\mathbf{W})] = n_{C,j}(\mathcal{S}_{\boldsymbol{\theta}},\mathbf{z}) \cdot (\mathbb{E}[C_C] + \mathbb{E}[C_E]) + n_{M,C,j}(\mathcal{S}_{\boldsymbol{\theta}},\mathbf{z}) \cdot \mathbb{E}[C_{M,Op}] \tag{41}$$

where $n_{C,j}(\mathcal{S}_{\boldsymbol{\theta}}, \mathbf{z})$ and $n_{M,C,j}(\mathcal{S}_{\boldsymbol{\theta}}, \mathbf{z})$ are the number of repair campaigns and component repairs in year $j$. Note the models defined in Eq. (39) and (41) do not explicitly account for the inspection and repair location and methods to simply the notation.

Based on Eq. (36), (39) and (40), the expected total lifetime cost can now be written to depend only on the heuristic I&M strategy $\mathcal{S}_{\boldsymbol{\theta}}$ and the inspection outcomes $\mathbf{Z} = \mathbf{z}$:

$$\mathbb{E}[C_2|\mathcal{S}_{\boldsymbol{\theta}}, \mathbf{z}] = \mathbb{E}[C_I|\mathcal{S}_{\boldsymbol{\theta}}, \mathbf{z}] + \mathbb{E}[C_M|\mathcal{S}_{\boldsymbol{\theta}}, \mathbf{z}] + \mathbb{E}[C_F|\mathcal{S}_{\boldsymbol{\theta}}, \mathbf{z}] \tag{42}$$

After evaluating the expected total lifetime cost conditional on the inspection outcomes $\mathbf{Z} = \mathbf{z}$, $\mathbb{E}[C_2|\mathcal{S}_{\boldsymbol{\theta}}]$ can be computed as:

$$\mathbb{E}[C_2|\mathcal{S}_{\boldsymbol{\theta}}] = \int_{\mathbf{Z}} \mathbb{E}[C_2|\mathcal{S}_{\boldsymbol{\theta}}, \mathbf{z}] \, p(\mathbf{z}|\mathcal{S}_{\boldsymbol{\theta}}) \, \mathrm{d}\mathbf{z} \tag{43}$$

The integral in Eq. (43) can be solved using a MC approach similar to Eq. (37).

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
