# Peer review of "Probabilistic cost modeling as a basis for optimizing inspection and maintenance of turbine support structures in offshore wind farms"

_Wind Energy Science, 2023_

## Author Comment (AC1)

The authors wish to thank the reviewers for their valuable time and effort they have devoted to reading the manuscript and providing constructive feedback. The authors have made every possible effort to address the comments and improve the manuscript.

Below, the authors' responses are given in *italics* together with the comments of the reviewers.

**Reviewer #1**

**Comment #1**: Line 41: "Typically, operation and maintenance (O&M) of an offshore wind farm corresponds to 25% - 30% of the levelized cost of energy (LCoE) (Ambuhl and Sorensen, 2017; Kolios and Brennan, 2018; Maples et al., 2013; Röckmann et al., 2017)."

Comment: Could you provide other references to confirm these values?

*Response #1: Thank you for pointing out the need to include further references to support the statement that "25% - 30% of the LCoE of an offshore wind farm can be attributed to the operation and maintenance activities". Please find below two relatively recent references, which support this statement.*

- *Stehly, T., Duffy, P., and Mulas, D.: 2022 Cost of Wind Energy Review, National Renewable Energy Laboratory, 2023*

- *OPEX Benchmark – An insight into operational expenditures of European offshore wind farms: PEAK Wind - Renewable Services, 2022. [https://peak-wind.com/update-2022-opex-benchmark-an-insight-into-the-operational-expenditures-of-european-offshore-wind-farms](https://peak-wind.com/update-2022-opex-benchmark-an-insight-into-the-operational-expenditures-of-european-offshore-wind-farms).*

*In addition to providing these extra references to support the above statement, we would like to point out that we have restructured and substantially revised the introduction of the manuscript to improve the presentation of the motivation for our work.*

*The revised introduction reads as follows (Section 1, line 32, page 2 - line 113, page 3):*

[revised manuscript text omitted]

Comment: Can you briefly comment on the other options? And how do they compare? Why specifically look at support structures? Is it just because it is a possible way? I think here, it is missing a more well-defined motivation.

***Response #2**: We agree that an optimization of I&M of the turbine support structures in offshore wind farms is not the only option to reduce the LCoE of an offshore wind farm. There are certainly other options. As an example, with regards to the wind turbines, actions aiming to increase the annual energy production (AEP) such as repowering can lead to a reduction in the LCoE of an offshore wind farm. Another efficient option for reducing the LCoE of an offshore wind farm is to extend its lifetime (see also Kinne, Farhan et al. 2022). To enable such a lifetime extension, the requirements regarding the reliability of the turbine support structures need to be fulfilled. This is typically only possible by applying suitable (monitoring-informed) I&M regimes for detecting and repairing deteriorated structural components. Such regimes can be optimized and adapted using decision-theoretical approaches as also outlined in our current contribution.*

*As highlighted in our response to Comment #1, we have decided to substantially revise the introduction to help the reader to better understand the motivation of our work. The revised introduction has already been reproduced in our reply to Comment #1.*

**Comment #3**: Figure 1. there is a mistake in the figure: some written content overlapping.

*Response #3: Thank you for pointing out the mistake in Figure 1. We have revised the figure accordingly.*

**Comment #4**: Comment on section 2: Can't you put some of the theory equations in an appendix, and translate all the math into a nice history? Readers can go to the appendix for more details if/when needed. In the way currently written, the whole history loses a bit of the flow due to the heavy math.

*Response #4: Thank you for your valuable advice on improving the readability of the manuscript. Following your advice, we have revised the presentation of the theory in Section 2 as well as in the numerical example in Section 6.2. In the revised draft, we have moved most of the demanding mathematics to Appendices A and B. As you suggested, the interested reader can now go to these appendices for more details on the computation of the expected utilities and costs required in the decision-theoretical optimization of I&M regimes.*

**Comment #5**: Section 7:

**5.1:** In my view, Equations 21-37 are all out of place. These should have been written in the methods section.

*Response #5.1: We very much appreciate your comment and have moved the equations to Appendix B (see also our response to Comment #4).*

**5.2:** The results are superficially discussed in Figures 12 and 13. Similar problems were identified for Tables 7 and 8, and the other figures in the section.

*Response #5.2: We thank the reviewer for his valuable advice and included a more elaborate discussion on the results contained in those figures and tables. Note that in the revised manuscript, Figure 12 is now Figure 10 and Figure 13 is now Figure 11.*

*The revised discussion on the results shown in Figure 10 is as follows (page 21, lines 602 - 613):*

*"Figure 10 shows the decomposed expected total lifetime cost $\mathbb{E}[C_2|\mathcal{S}_{\theta}]$ in function of the heuristic parameters $\boldsymbol{\theta} = [\Delta t = 8, p_{th} = 1 \cdot 10^{-3}, n_{I,C} = 1, ...,22, \eta = 1, a_R = 1]^T$. This cost is composed of the expected values of the failure cost, inspection campaign cost, inspection operation cost, repair campaign cost, repair operation cost, and engineering cost for repairs. As the inspection effort increases (i.e., more hotspots are inspected during each inspection campaign), the expected value of the system failure cost (i.e., the risk of structural failure) decreases, and – as expected – the expected values of the inspection and repair costs increase. This nicely illustrates the impact of the risk mitigation measures on the structural risk of failure. Note that the engineering cost for repairs is constant in this case study since it is here incurred only once at the beginning of the operational phase as repair solutions are engineered proactively before the frame is commissioned. Consequently, they could be neglected in the current optimization, as they only shift the expected total lifetime costs upwards by a fixed value."*

*The revised discussion on the results shown in Figure 11 is as follows (page 22, lines 617 - 627):*

*"To support the decision on whether one should implement a I&M strategy, the predicted value of information and actions is computed by the difference between the expected total lifetime cost $\mathbb{E}[C_0]$ and the expected total lifetime cost $\mathbb{E}[C_2|\mathcal{S}_\theta]$. By normalizing this difference with respect to $\mathbb{E}[C_0]$, the relative $\bar{V}_{SS-A}^{PIPA-DA}(\boldsymbol{\theta})$ is obtained (Farhan, Schneider and Thöns, 2021):*

$$\bar{V}_{SS-A}^{PIPA-DA}(\boldsymbol{\theta}) = \frac{\mathbb{E}[C_0] - \mathbb{E}[C_2|\mathcal{S}_\theta]}{\mathbb{E}[C_0]} \qquad (1)$$

*Figure 11 shows the $\overline{VoI}_{SS-A}^{PIPA-DA}$ in function of the parameters $\boldsymbol{\theta} = [\Delta t = 8, p_{th} = 1 \cdot 10^{-3}, n_{I,C} = 1, \dots, 22, \eta = 1, a_R = 1]^T$, where $\Delta t = 8$ and $p_{th} = 1 \cdot 10^{-3}$ are the optimal inspection interval and reliability threshold. The dashed blue line corresponds to the expected total lifetime cost $\mathbb{E}[C_0]$ determined by the SS-A. The dashed-dotted blue line corresponds to the expected total lifetime cost $\mathbb{E}[C_2|\mathcal{S}_\theta]$. Notably, $\overline{VoI}_{SS-A}^{PIPA-DA}$ is positive for $n_{I,C} = 1, \dots, 22$. This result indicates that it is a rational decision to inspect and maintain the frame. As expected, the highest $\overline{VoI}_{SS-A}^{PIPA-DA}$ is obtained when implementing the optimal strategy $\mathcal{S}_{\theta^*}$ with an optimal number of inspected hotspots in each inspection campaign $n_{I,C} = 6$."*

*The revised discussion on the results summarized in Table 7 and 8 reads as follows (page 24, lines 649 - 655):*

*"Aligning with existing works, we utilize the probabilistic I&M cost model to derive deterministic normalized cost models. In the literature, such a normalization is typically performed with respect to the failure cost or campaign cost due to their significant contribution to the overall lifetime costs. Applying the same methodology, we obtain the normalized cost models summarized in Table 7 and Table 8. These models can subsequently be used to optimize I&M of offshore wind turbine support structures if the costs are included in the underlying models on a linear basis."*

**5.3:** In general, this section lacks a more objective and comprehensive discussion of the numerical results for the example.

*__Response #5.3__: We thank for the reviewer for his valuable advice and improved the discussion on the results of the numerical example. In addition to the discussion of the results in Figures 10 and 11 as well as Tables 7 and 8 (see our reply to Comment #5.2), we included the following discussion on the results in Figure 9 (page 21, lines 589 – 598):*

*"[...] The estimated expected total lifetime cost $\mathbb{E}[C_2|\mathcal{S}_\theta]$ in function of $\boldsymbol{\theta}$ is shown in Figure 9.*

*All strategies with $n_{I,C} = \{3,4,5,6\}$ result in a similar expected total lifetime cost. This provides some flexibility to the decision-maker to choose a strategy based on their specific requirements regarding the inspection interval and structural reliability. Notably, both strategies with $\Delta t = 4$ years, exhibit similar expected costs for $n_{I,C} > 7$ regardless of the*

*reliability threshold. When considering strategies with $\Delta t = 8$, the reliability threshold has an impact on the expected total lifetime cost: a lower threshold results in more unscheduled inspections between regular inspections. In our current example the optimal strategy $\mathcal{S}_{\boldsymbol{\theta}^*}$ is characterized by $\boldsymbol{\theta}^* = [\Delta t = 8, p_{th} = 1 \cdot 10^{-3}, n_{I,C} = 6, \eta = 1, a_R = 1]^T$."*

*The revised discussion on the results in Figure 12 is as follows (page 23, lines 640 - 646):*

*"[...] The estimated expected total lifetime cost $\mathbb{E}[C_2|\mathcal{S}_{\boldsymbol{\theta}}]$ considering expected I&M costs are shown in Figure 12.*

*It can be seen that the current analysis provides the same results as the analysis considering the probabilistic I&M costs (cf. also Figure 9) and thus the same optimal strategy $\mathcal{S}_{\boldsymbol{\theta}^*}$ with $\boldsymbol{\theta}^* = [\Delta t = 8, p_{th} = 1 \cdot 10^{-3}, n_{I,C} = 6, \eta = 1, a_R = 1]^T$. Consequently, the current analysis illustrates that the I&M costs can be considered deterministically as expected values in the decision and VoI analysis if they are included in the optimization on a linear basis."*

**Comment #6:** Your work assumed a lognormal distribution for the majority of your statistical fits. I am missing some sort of discussion on the implications of these assumptions. Why not another distribution? Could your results be changed otherwise?

> ***Response #6:*** *We very much appreciate your comment and acknowledge that a more detailed discussion on our choice of the probabilistic models of the parameters of the cost model would help the reader to understand our reasoning behind this choice. Therefore, we have revised the corresponding discussion as follows (page 10, lines 335 - 363):*
>
> *"Given the lack of empirical data on the uncertain parameters of the cost models $\boldsymbol{W}$, their probabilistic models are – in a Bayesian sense – chosen based on the available expert knowledge. It should be emphasized upfront, however, that these probabilistic models can be updated using Bayesian methods if data on the parameters $\boldsymbol{W}$ become available.*
>
> *As a first step in the probabilistic model building, the parameters in $\boldsymbol{W}$ are assumed to be independent and their marginal distributions are assumed to follow the lognormal distribution. The first assumption is made as no information on the correlation of the different parameters is available. The second assumption is supported by the following reasons. First, each parameter of the cost models only takes non-negative values, and their statistical distribution is typically unimodal, i.e., one range of values in the distribution occurs more frequently than other ranges of values. The lognormal distribution is commonly chosen to probabilistically model such quantities as it is bounded by zero, has no upper limit, and is unimodal. Second, the lognormal distribution is skewed to the right with a long tail capturing rare extreme values of the cost model parameters. Third, the assumption that the parameters are lognormal distributed can be partially explained by the central limit theorem. Assuming that each parameter of the cost model itself derives from a multiplicative process, the sum of the logarithms of the factors in the underlying process approaches a normal distribution and their product approaches a lognormal distribution as the number of factors becomes large. For these reasons, the lognormal distribution is a plausible probabilistic model for the different cost model parameters (see also Moy, Chen and Kao. 2015).*
>
> *As a second step in the model building, the statistics of the different lognormal distributions are determined based on the lower and upper bounds of each cost model parameter*

*specified in Section 3.2. These bounds represent the available expert knowledge on the ranges of the parameter values. Based on additional expert judgement, the lower and upper bound are assumed to characterize the 1% and 95%-quantile of the parameter values. Using this information, the different lognormal distributions are fitted as illustrated in Figure 2. The resulting mean and coefficient of variation (CoV) of each probabilistic parameter of the cost models is summarized Table 5."*

**Comment #7**: The model is not completely reproducible, as some of the assumptions are based on interviews with project owners and stakeholders. There is little information about it in the article. Therefore, I would like to see further details on the assumptions as much as possible. What would be the implications of not having these pieces of information? Could you have done your work without it? How would it be affected? I am missing a discussion here.

> ***Response #7****: We thank the reviewer for his valuable comment. However, we would like to point out that the probabilistic model of the cost model parameters can be reproduced with the information provided in Sections 3 and 4. First, based on the interviews, we established the parameters influencing the I&M costs as described in Section 3.2. In the same section, we documented the ranges of the parameter values, which we also established based on the interviews. Subsequently, in Section 4.2, we outline the deterministic model for estimating the total I&M costs. These models capture the level of detail and the operational constraints, which we were able to establish based on the interviews. Finally, in Section 4.2, we discuss the process of building the probabilistic model of the parameters of the cost model. In the revised manuscript, we provide a detailed discussion on how we derived the marginal distributions of the model parameters based on the parameter bounds provided by the interviewed experts (see also our reply to Comment #6). The same methodology can be applied to establish I&M cost models for other types of wind farms with different operational and logistical constraints provided that the required level of expert knowledge is available. Ideally, detailed documentation of the operational constraints and logistical procedures as well as empirical data on the parameters governing the I&M activities and costs can be provided as a basis of the model building.*

**Reviewer #2**

**Comment #1**: The second sentence of the Introduction "The integrity management..." is awkward and I don't understand what it is trying to say. There are a handful of other sentences and phrases in the manuscript like this to. Perhaps a professional editor could help.

> ***Response #1****: We very much appreciate your comment. As a result, we have revised a significant portion of the manuscript to improve the presentation of the material. All changes to the manuscript are highlighted in green color. Note that we have restructured and substantially revised the introduction of the manuscript to improve the presentation of the motivation for our contribution. Please refer to Section 1, line 32 on pages 2 to line 113 on page 3 in the revised manuscript.*

**Comment #2**: The fourth paragraph of the Introduction (around line 60 on page 2) throws 15 citations at the reader in one shot. I think the authors could go a bit further to describe this previous work and similarities or distinctions between them.

***Response #2:*** *Thank you for your comment. As described in our reply to Comment #1, we have revised the introduction to discuss our work in relation to existing works. The relevant part of the introduction reads as follows (line 55 on page 2 to line 88 on page 3):*

*"Clearly, I&M of deteriorating WT support structures in an offshore wind farm is associated with costs and the total lifetime I&M costs depend on the adopted I&M strategy, which determines the time and scope of each I&M campaign based on the available system information. Condition-based and predictive maintenance strategies can be optimized at the beginning of and adapted during the planned and/or extended lifetime of an offshore wind farm using preposterior analysis from Bayesian decision theory (e.g., Sorensen, 2009; Nielsen and Sorensen, 2011; Florian and Sorensen, 2017; Farhan, Schneider and Thöns, 2021; Bismut and Straub, 2021). In such an analysis, probabilistic models of (a) the governing deterioration processes including the effect of maintenance, (b) the structural performance, and (c) the inspection/monitoring performance are employed to predict:*

- *the condition of the structural components, inspection/monitoring outcomes and maintenance actions, and*

- *the structural component/system reliability conditional on the predicted component condition, inspection/monitoring outcomes, and maintenance actions.*

*In addition, a cost model is utilized to quantify the costs of inspections/monitoring and maintenance as well as the monetarized consequences of structural failures. Based on these models, the expected lifetime I&M costs and the lifetime risk of structural failure can be estimated for a given I&M strategy. A cost and risk optimal I&M strategy then balances the expected lifetime I&M costs with the lifetime risk of structural failure.*

*In the existing literature, normalized cost ratios or deterministic cost models are utilized as a basis for optimizing I&M of deteriorating structural systems using decision-theoretical approaches (e.g., Schneider, Rogge, Thöns et al., 2018; Bismut and Straub, 2021; Morato, Andriotis, Papakonstantinou et al., 2023). Although deterministic cost models enable an optimization of I&M activities, they lack the ability to capture the effect of the I&M cost uncertainties in the decision analysis; especially in applications in which I&M costs are included in the underlying models on a non-linear basis. Importantly, probabilistic parametric cost modeling facilitates sensitivity analyses (beyond local derivative-based sensitivity analyses) to understand the effect of the various uncertain cost-affecting factors on the total I&M costs. With regards to optimizing I&M of WT support structures in offshore wind farms, comprehensive and explicit consideration of probabilistic I&M costs in the decision analysis is – to the best of the authors' knowledge – something which has not been explored previously. In addition, this issue is also relevant, since – from our experience – wind farm operators commonly highlight the need to consider the uncertainties in the I&M costs in the optimization of I&M strategies."*

**Comment #3**: - Introduction line 75, "In the numerical example, we demonstrate that the I&M costs can be considered deterministically as expected values in the analysis since they are included in the model on a linear basis." This statement and conclusion do carry through the paper from beginning to end, but it also undercuts the approach and leaves me unsure of its contributions. The authors note previous research where deterministic models were used probabilistically, so what is

the meaningful difference? The authors should be clearer about what is new and novel in this work here. More thoughts on this theme in later comments as well.

> ***Response #3:*** *We really appreciate your comment and have removed this conclusion from the abstract and introduction to better focus on what is new and novel in our work. Nevertheless, as demonstrated in the numerical example, if the I&M costs enter the models underlying the optimization on a linear basis, it is possible to simplify the cost model by considering the expected value of the cost model parameters. However, as you rightly highlight, one of the main advantages of the I&M cost using a probabilistic parametric cost model is that it allows us to perform a sensitivity analysis to identifying the main cost drivers and their impact on I&M decisions.*

**Comment #4:** Figure 1: There is overlapping text in the PDF in the top bar header of the figure.

> ***Response # 1:*** *We thank the reviewer for pointing out the mistake in Figure 1. We have corrected it accordingly.*

**Comment #5:** Tables 1, 2, 3, 4: Where are these input values coming from? Data? Interviews? Modeled processes? Data for operations and maintenance is the hardest to come by, so the authors need to be precise when reporting their inputs to their model. I suggest citations in the caption.

> ***Response #5:*** *We very much appreciate your comment and agree that it is a challenge to establish data and information the cost of I&M of support structures in offshore wind farms. As a basis of our work, we performed interviews with our project partners, which are not publicly available for citation. Instead, we have documented the methodology, which we applied to derive the models presented in our contributions. This methodology can be summarized as follows: The basic logistical procedures applied by our project partners (utilization of workboats operating from a port base to conduct separate inspection and repair campaigns) formed the very basis for deriving the parametric I&M cost model. We conducted interviews with a wind farm operator, engineering consultants, and O&M engineers. These experts provided information, which allowed us to identify the main parameters influencing the total I&M costs in addition to estimates of the bounds on the parameter values. Given the level of detail that could be established from the interviews, we first established the parametric model for describing the total I&M costs. Subsequently, we derived a probabilistic model of the cost model parameters based on the parameter bounds provide by the experts (see also our reply to Comment #6).*

**Comment #6:** - Page 10, "Due to the lack of data on the parameters ... their marginal probability distributions are assumed to be lognormal." Where does this assumption come from. With just a min and a max and a lack of data, seems like a uniform distribution, or maybe even a triangular distribution with a "most likely value" would be better approximations in a sparse data context. Why lognormal? Why not normal or any other distribution? This seems to be a critical assumption because the output distributions are also lognormal, so they track the input assumptions closely.

> ***Response #6:*** *We very much appreciate your comment and acknowledge that a more detailed discussion on our choice of the probabilistic models of the parameters of the cost model would help the reader to follow the material presented in our manuscript. Therefore, we have revised the corresponding discussion as follows (page 10, lines 335 - 363):*

> *"Given the lack of empirical data on the uncertain parameters of the cost models **W**, their probabilistic models are – in a Bayesian sense –chosen based on the available expert*

*knowledge. It should be emphasized upfront, however, that these probabilistic models can be updated using Bayesian methods if data on the parameters **W** become available.*

*As a first step in the probabilistic model building, the parameters in **W** are assumed to be independent and their marginal distributions are assumed to follow the lognormal distribution. The first assumption is made as no information on the correlation of the different parameters is available. The second assumption is supported by the following reasons. First, each parameter of the cost models only takes non-negative values, and their statistical distribution is typically unimodal, i.e., one value in the distribution occurs more frequently than other values. The lognormal distribution is commonly chosen to probabilistically model such quantities as it is bounded by zero, has no upper limit, and unimodal. Second, the lognormal distribution is skewed to the right with a long tail capturing rare extreme values of the cost model parameters. Third, the assumption that the parameters are lognormal distributed can be partially explained by the central limit theorem. Assuming that each parameter of the cost model itself derives from a multiplicative process, the sum of the logarithms of the factors in the underlying process approaches a normal distribution and their product approaches a lognormal distribution as the number of factors becomes large. For these reasons, the lognormal distribution is a plausible probabilistic model for the different cost model parameters (see also Moy, Chen and Kao, 2015).*

*As a second step in the model building, the statistics of the different lognormal distributions are determined based on the lower and upper bounds of each cost model parameter specified in Section 3.2. These bounds represent the available expert knowledge on the ranges of the parameter values. Based on additional expert judgement, the lower and upper bound are assumed to characterize the 1% and 95%-quantile of the parameter values. Using this information, the different lognormal distributions are fitted as illustrated in Figure 2. The resulting mean and coefficient of variation (CoV) of each probabilistic parameter of the cost models is summarized Table 5."*

**Comment #7:** - Table 5: I am confused by this table. I understood how Tables 1-4 are converted to be lognormal based on min & max values, but where do these distributions come from? Are these inputs or outputs? If inputs, then sources and citations must be included. Also, how would a parameter such as "Campaign cost" be an input when Section 4 described the elements that comprise a "campaign"

*Response #7: Table 5 summarizes the probabilistic models of the input parameters of the parametric cost model defined in Eq. (11) and (12). These models are established as described in our replies to Comment #5 and #6. To summarize, lognormal distributions were chosen based on the discussion described in our reply to Comment #6. The parameters of the lognormal distributions were determined based on the parameter bounds provided in Tables 1 to 4, which we established through expert interviews. Based on the additional expert judgment that 1% of the parameter values are smaller than lower bound and 5% of the parameter are larger than upper bound, we determined the statistics of the lognormal distributions as documented in Table 5.*

*Furthermore, as explained in the revised manuscript (page, lines 295 - 298, the campaign cost corresponds to the fixed one-time cost of initiating the I&M activities, which includes the cost of commuting to the wind farm and back, the cost of the equipment required for*

*the planned activities, fuel costs, and project management costs. These cost components are included in the mobilization and demobilization cost of the vessels).*

*The campaign costs are one component of the total I&M costs. The information we retrieved from the interviews did not allow us to explicitly model the factors influencing the campaign cost in more detail. Instead, we have summarized this cost component in a single cost model parameter. We were, however, able to establish bounds on this cost component, which are expressed as mobilization and demobilization cost in Table 1. The proposed model thus reflects the level of detail we were able to gather from the interviews and expert opinion.*

*Note that this is not a limitation, as the proposed cost model can be extended to account for a more refined model for describing the campaign cost if additional details on this cost component can be established.*

**Comment #8:** - Figures 3-5: The outputs are clearly lognormal to match the input distributions, which asks the question of what is "probabilistic" about this model if all of the operations are linear enough to maintain the input distribution properties. Phrased another way- what is gained by using probabilistic inputs vs deterministic inputs here. Alternatively, if the authors had assumed uniform input distributions, would their conclusions be any different?

*Response #8: Thank you for your valuable comment. To better clarify the value of using a probabilistic approach to modelling I&M costs, we revised the discussion of the results presented in Figures 3 to 5 as follows (page 15, lines 411 – 421):*

*"From Figure 3 to Figure 5, it can be seen that propagating the uncertainties in the cost model parameters through the cost models defined in Eq. (11) and (12) provides a probabilistic description of the total I&M costs. In each considered scenario, the total I&M costs exhibit an approximate lognormal distribution. The statistics of the total I&M costs shown Figure 3 to Figure 5 are summarized in Table 6. Note that the coefficients of variation (CoV) of the total I&M costs indicate that in each scenario certain parameters dominate the uncertainty in the total I&M costs. As an example, in Figure 3a (wind farm level analysis considering EM inspection), the CoV of the total I&M costs is similar to the CoV of the vessel cost per shift. This finding is further substantiated by the variance-based sensitivity analysis in Section 5.2.1, where in Figure 6 we observe that the vessel cost per shift has a sensitivity index close to one for the same inspection scenario."*

*One of the main advantages of utilizing probabilistic parametrized cost models to describe the total I&M costs is that such an approach enables sensitivity analyses to establish the main cost drivers. We highlighted this advantage in the introduction of the revised manuscript as follows (page 3, lines 81 - 83):*

*"[…] Importantly, probabilistic parametric cost modeling facilitates sensitivity analyses (beyond local derivative-based sensitivity analyses) to understand the effect of the various uncertain cost-affecting factors on the total I&M costs."*

*It is important to highlight that it is possible to optimize I&M activities using deterministic cost models. However, as highlighted in the introduction of the revised manuscript (page 2, line 77 to page 3, line 80), it is also important to understand that "[…] although deterministic cost models enable an optimization of I&M activities, they lack the ability to*

*fully capture the effect of the uncertainties in the I&M costs in the decision analysis especially in applications in which I&M costs are included in the underlying models on a non-linear basis. [...]".*

*To the best of our knowledge, our contribution for the first time considers consistently and explicitly uncertainties in the I&M costs in the decision-theoretical optimization of I&M regimes through the application of probabilistic parametric I&M cost models. As highlighted in the revised introduction (page 3, lines 86 – 88), "[...] this issue is also relevant, since – from our experience – wind farm operators commonly highlight the need to consider the uncertainties in the I&M costs in the optimization of I&M strategies."*

*Finally, we discussed in more detail in our revised manuscript (page 10, lines 335 - 363) why the lognormal distribution is a plausible probabilistic model for the different cost model parameters (see also our reply to Comment #6).*

**Comment #9:** Section 6 Sensitivity Analysis: I applaud the authors for including a sensitivity analysis, as that is typically where probabilistic analysis gives the greatest insight. However, I am struggling with the communication of results in Figures 6-9. It seems like the authors are trying to show too many dimensions of variation all at once and the important comparisons are spread out across the figures. Additionally, the figures are small and tough to read or see. Also, in the end, only three variables are compared at one time in the different color lines. The impact of the sensitivity analysis is to compare the relative impact of many variables all at once. Why not combine Figures 6-9 and all of the above/below water differences, show more comparisons, perhaps with a bar chart, and then show some perturbations, such as a bar chart with 1 or 5 inspected components. The way the sensitivity analysis is presented now, I have little meaningful takeaways or insights into the problem.

> *Response #9: We appreciate the reviewer's insightful comments and valuable advice on improving this section. As a result, we have reduced the number of figures in Section 5 (previously Section 6) and revised the content of the remaining figures to reduce the information presented in each figure. These adjustments enhance the readability and facilitate a clearer interpretation of the results, while still supporting our discussion and conclusions.*

**Comment #10:** Figure 10: I don't fully understand the physics that are driving the failure model of the jacket. Is there is a time-series load signal from a turbine that is propagated to the jacket? Something more prescribed or analytical? Are all of the 22-stress concentration hot spots at the welded joints considered independent random variables? If so, how realistic is that? It seems as though the later analysis on the optimal number of inspections is highly dependent on this assumption, because of the tendency to inspect all of them as soon as one crack is found. Furthermore, there is a greater likelihood to continue to inspect all of them after a crack is found and repaired.

> *Response # 10: Thank you for your valuable comment. We appreciate that a more detailed description of the physics governing the behavior of frame and the probabilistic models describing the stochastic dependencies among fatigue behavior of the hotspots improves the manuscript. To this end, we revised the description of the frame as follows (page 17, line 484 to page 18, line 512)*

*"[…] The steel frame is made of welded tubular sections. Its planned lifetime is 25 years, which is divided into $j = 1, \dots, m$ intervals of one year length. During the operational phase, the frame is exposed to a time-dependent lateral force representing a storm load. This load is modeled by its annual maximum $L_{max,j}$. In addition to storm loads, the frame – like a jacket support structure of an offshore wind turbine – is subject to fatigue due to dynamic excitations. In the current analysis, the welded connections of the frame contain 22 critical fatigue hotspots, which are indicated as red dots in Figure 8. The hotspot fatigue demand is quantified by the corresponding distributions of the fatigue stress ranges. Typically, these distributions are derived from an overall dynamic response analysis. In the current example, they are – as described in Straub (2004) – determined based on the available design information (i.e. the hotspot fatigue lives and the applied SN curves).*

*Fatigue deterioration of the hotspots is described by probabilistic Paris-Erdogan fatigue crack growth models. The statistical dependence among the fatigue behavior of different hotspots is captured by introducing correlations among the uncertain parameters of the hotspot fatigue models. This correlation influences the system reliability and has an impact on the (optimal) I&M regime.*

*The hotspots are inspected with MPI via a CTV and repaired by welding if required. The applied repair model is documented in detail in (Farhan, Schneider et al. 2021). It is assumed that hotspots 1 to 8 are located above water, while hotspots 9 to 22 are located below water. The location of the of the hotspots (above or below water) influences the cost of inspections and repairs.*

*The time-dependent failure probability is computed by coupling the probabilistic fatigue deterioration models with a probabilistic structural performance model utilized to evaluate the system failure probability conditional on the hotspot condition. Inspection information is included in the estimation of the system failure probability through Bayesian updating of the probabilistic fatigue models. Further information regarding the applied fatigue, structural performance, and inspection models as well as the methods employed to compute the (updated) time-variant failure probability of the frame is documented in (Schneider, Thöns et al. 2017, Schneider 2020, Eichner, Schneider et al. 2023). […]"*

*In addition, we added the following discussion (page 20, lines 543 - 566) on the impact of*

- *the rules guiding the decisions on inspection and repair, and*
- *the stochastic dependence among the fatigue behavior of the hotspots*

*on the inspection and repair effort.*

*"[…] In the current application, the parameterized rules are defined as follows (see also Bismut, Luque et al. 2017, Schneider 2019, Eichner, Schneider et al. 2023):*

1. *Inspection campaigns are performed at fixed intervals $\Delta t$.*

2. *$n_{I,C}$ hotspots are inspected during each inspection campaign.*

3. *Hotspots are prioritized for inspection according to a metric proposed by Bismut, Luque et al. (2017), which is a function of a parameter $\eta$ as well as the structural importance and fatigue reliability of each hotspot.*

4. *An additional inspection campaign is launched if the predicted annual system failure probability exceeds a threshold $p_{th}$.*

5. *A maintenance campaign is launched if fatigue cracks are indicated and measured to be deeper than $a_R$.*

*Note that rules 4 and 5 have the following implication: Inspection information obtained at one hotspot contains indirect information on the fatigue state of the remaining hotspots as their fatigue behavior is correlated due to common influencing factors (see Straub and Faber 2004 for a detailed discussion). Consider now the case in which a fatigue crack is unexpectedly indicated at one hotspot and measured to be deeper than $a_R$. Conditional on this inspection information, the probability that fatigue deterioration of the remaining hotspots has progressed faster than expected increases. Consequently, the system failure probability also increases. If it exceeds the threshold $p_{th}$, additional inspections and possibly repairs are performed as prescribed by rules 4 and 5. Because of these two rules and the explicit modeling of the dependence among the fatigue behavior of different hotspots, the current optimization of I&M of the frame captures scenarios in which the inspection and repair effort has to be increased due to accelerated fatigue deterioration. [...]"*

**Comment #11:** Section 7.2: The math on the first couple of pages here is probably best moved to an Appendix. Once the Appendix is created though, the authors should consider shunting other equations there as well to improve the paper's readability for the wind-focused audience of the journal.

> ***Response # 11:** We very much appreciate your advice on improving the readability of the manuscript. As a result, we have revised the presentation of the theory in Section 2 as well as in the numerical example in Section 6.2. In the revised draft, we have moved most of the demanding mathematics to Appendices A and B.*

**Comment #12:** Section 7.3: I don't follow what is different about this section and the results. Figure 14 looks the same as Figure 11.

> ***Response # 12:** Thank you for your comment. In Section 6.3 (previously Section 7.3), we explore the potential for replacing the probabilistic cost model with a deterministic one, which utilizes the expected values of the cost model parameters. In addition, we discuss in what situations such a simplification is appropriate. This is the case if the costs enter the model on a linear basis. To better highlight our intention, we have revised the discussion on the results in Section 6.3 as follows (page 23, lines 640 - 646):*

> *"The estimated expected total lifetime cost $\mathbb{E}[C_2|\mathcal{S}_\theta]$ considering expected I&M costs are shown in Figure 12.*

> *It can be seen that the current analysis provides the same results as the analysis considering the probabilistic I&M costs (cf. also Figure 9) and thus the same optimal strategy $\mathcal{S}_{\theta^*}$ with $\theta^* = [\Delta t = 8, p_{th} = 1 \cdot 10^{-3}, n_{I,C} = 6, \eta = 1, a_R = 1]^T$. Consequently, the current analysis illustrates that the I&M costs can be considered deterministically as expected values in the decision and VoI analysis if they are included in the optimization on a linear basis."*